# Gecko: A Simulation Environment with Stateful Feedback for Refining Agent Tool Calls

**Zeyu Zhang** [1][2][3]   **Guohao Li** [2][3]   **Zhenchang Xing** [4]   **Alexandros Apostolopoulos** [5]   **Yu Lin Lee** [5]   **Liang Zheng** [1]

## Abstract

The ability to use tools is fundamental for large language model (LLM) agents. Given a task, existing systems use LLMs to plan and generate tool calls, which are executed by real-world tools to complete the task. However, tool calls are prone to errors because they are generated primarily from the intrinsic capabilities of LLMs. Moreover, while it is useful to let LLMs iteratively refine the tool-call sequence using execution results from real tools, this process can be expensive and may cause unsafe side effects. To improve LLM tool calls and address issues caused by using real tools for refinement, we introduce Gecko[1], a stateful simulation environment that provides informative feedback for refining LLM tool calls before real execution. Specifically, Gecko combines rules and LLMs to check the validity of tool names and arguments, synthesize schema-conforming and state-consistent responses, and judge task completion against the user objective. These three types of feedback allow LLMs to refine their tool calls in simulation, forming a simple yet effective test-time scaling method named GATS. On BFCLv3 and $\tau^2$-bench, GATS consistently improves the performance of various LLMs, including GPT-4o, GPT-5, and Gemini-3.0-pro (Fig. 1).

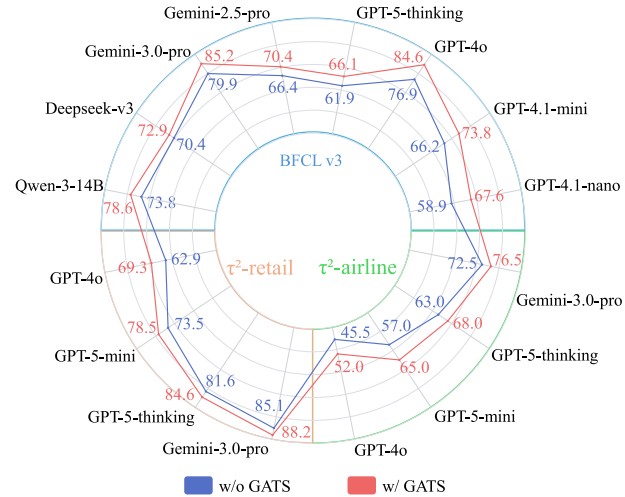

*Figure 1.* GATS uses Gecko's feedback from simulated tool executions to iteratively refine tool calls, consistently improving the performance of different LLMs on BFCLv3 and $\tau^2$-bench.

## 1. Introduction

Building agent systems using LLMs to solve complex tasks has become increasingly popular. In this mission, it is critical to let LLMs be able to use external tools, such as `get_weather` and `fetch_stock_data`. While

there exist strong LLMs such as GPT-4o (OpenAI, 2024b), Qwen3 (Yang et al., 2025), and xLAM-2 (Prabhakar et al., 2025), because of long contexts, high task complexity, and rigid tool definitions, it is still challenging for these LLMs to select suitable tools and provide accurate arguments (Kate et al., 2025; Huang et al., 2024).

To improve the tool-calling capabilities of LLMs, some existing methods let LLMs use real tools and use the execution results as feedback to refine tool-use decisions (Singh et al., 2025; Shi et al., 2024; Kang et al., 2025; Li et al., 2025). While this strategy can help, repeatedly using real tools for refinement can be costly and risky. For example, real APIs may incur service fees or rate limits (*e.g.*, RapidAPI), and state-changing tools may produce undesirable external side effects (Li & Fung, 2025). An inappropriate call to `Tweet_Post` may publish unintended or private content, while erroneous calls to tools such as file deletion, email sending, or order placement may be difficult to safely undo.

In this work, our goal is to improve LLM tool-call performance while avoiding real-tool trial-and-error during test-time refinement. We refer to the LLM that proposes tool-call sequences during simulated refinement as the *planning LLM*.

---

[1]Australian National University, Australia [2]CAMEL-AI.org [3]Eigent.AI, United Kingdom [4]CSIRO's Data61, Australia [5]Aipotheosis Labs, United Kingdom. Correspondence to: Liang Zheng <liang.zheng@anu.edu.au>.

*Proceedings of the 43rd International Conference on Machine Learning*, Seoul, South Korea. PMLR 306, 2026. Copyright 2026 by the author(s).

[1]Gecko comes from keywords a**ge**nt + feedba**ck** + envi**ro**nment.

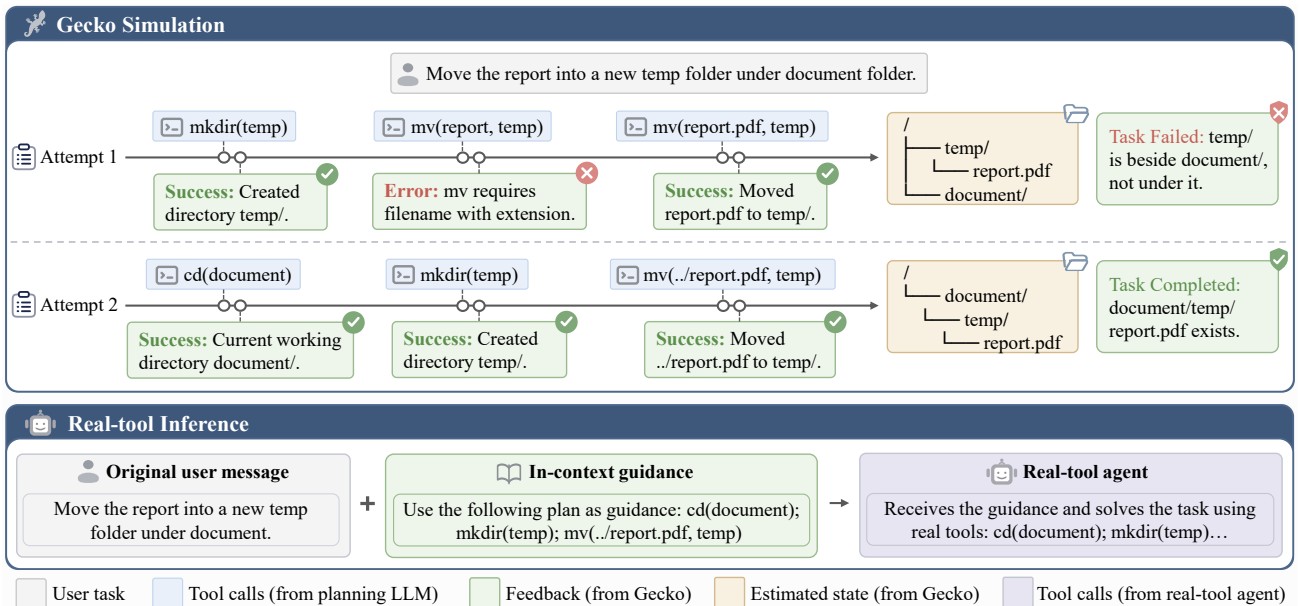

*Figure 2.* Overview of Gecko simulation and real-tool inference. Given a user task, a planning LLM first explores in Gecko. For each attempt, Gecko validates tool calls, simulates tool responses, updates an estimated task state, and judges whether the task is failed or completed by comparing the estimated state with the user objective. In this example, Gecko marks the first attempt as failed because the estimated state shows that `temp/` is beside `document/` rather than under it and marks the second attempt as completed. The successful plan is then prepended to the original user message as in-context guidance, and the real-tool agent solves the task using real tools.

To this end, we introduce Gecko, a stateful simulation environment that simulates tools from their schemas, follows the same input and output formats as real tools, and produces semantically plausible responses. Given a user task and candidate tool calls from the planning LLM, Gecko grounds tool calls through three types of feedback. **First**, Gecko validates tool calls, including tool names and input arguments, *e.g.*, "input 2 is invalid because only float numbers are allowed." This validation is implemented by a combination of rule-based checks and a helper LLM. **Second**, Gecko simulates schema-conforming tool responses. To make these responses task-relevant and consistent with prior tool calls, Gecko prompts a helper LLM with the validated tool call, the tool schema, and the current task state. **Third**, Gecko updates an estimated task state that reflects task progress, and a judge LLM compares this state with the user objective to determine whether the attempt is failed or completed.

Building on this feedback, we design Grounding Agent Test-time Scaling (GATS), a refinement method that lets the planning LLM explore candidate tool-call plans in Gecko before real execution. As shown in Fig. 2, each failed simulated attempt provides feedback for the next attempt, while a completed simulated plan is prepended to the original user message as in-context guidance. The real-tool agent then solves the task using real tools, guided by the successful simulated plan. In this way, GATS allows LLM agents to preview likely execution outcomes and learn from failed attempts in a virtual environment, without carrying their costs or side effects into the real environment.

We perform extensive evaluation on the BFCLv3 (Patil et al., 2025) and $\tau^2$-bench (Barres et al., 2025), where Gecko instantiates simulated versions of 8,578 and 13 benchmark tools from their specifications, respectively. We show that tool calls generated by many LLMs, such as GPT-4o (OpenAI, 2024b) and GPT-5 (OpenAI, 2025b), can be effectively hosted and executed in Gecko. As shown in Fig. 1 and Table 3, improvements are consistent across different planning LLMs and across both single-turn and multi-turn tasks. For example, the overall performance of GPT-4o is improved from 76.93% to 84.62% on BFCLv3.

To summarize, this paper makes the following contributions:

- We introduce Gecko, a stateful simulation environment that provides validation, simulated tool responses, and task-level feedback for pre-execution refinement of LLM tool calls.

- We propose GATS, a test-time scaling method that uses Gecko as a simulation sandbox to iteratively refine tool-call trajectories before final execution with real tools.

- We show that GATS brings consistent improvement to existing LLMs on BFCLv3 and $\tau^2$-bench.

- We analyze the mechanisms, limitations, and future directions made possible by Gecko.

## 2. Related Work

**Improving LLMs' intrinsic tool-calling abilities.** ToolAlpaca (Tang et al., 2023) fine-tunes LLMs on tool-use data generated by strong teacher models like GPT-4 (OpenAI, 2024a). ToolLLM (Qin et al., 2024) collects a large number of real-world APIs and uses an automatic pipeline to construct instruction-tuning data for tool-use fine-tuning. API-Gen (Liu et al., 2024c) and ToolACE (Liu et al., 2024a) improve the quality of synthetic tool-use data by format checking and semantic verification to improve fine-tuning. These methods mainly focus on training data synthesis. In comparison, Gecko naturally supports test-time scaling. Gecko also has strong potential in data synthesis and reinforcement learning (refer to Section 5).

**Test-time scaling for agentic tool use.** Existing methods use feedback loops or self-reflection grounded in *real* tool execution (Shi et al., 2024; Du et al., 2024; Qiao et al., 2024; Singh et al., 2025; Li et al., 2025; Chen et al., 2025b; Shi et al., 2025; Shinn et al., 2023; Zhou et al., 2025). For example, ConAgent (Shi et al., 2024) iteratively refines tool calls using feedback generated by an observation LLM from real tool failure messages. TRICE (Qiao et al., 2024) combines behaviour cloning with reinforcement learning guided by real tool execution feedback, teaching the model to refine its tool calls. These methods rely on repeatedly calling real tools, leading to tool-call costs and potential side effects. In contrast, Gecko removes the need for real tool executions in test-time scaling. Moreover, while these methods provide feedback on correcting individual failed tool calls, without maintaining a task state, they are unable to provide task-level feedback.

**Simulation environments for agentic tool use.** Existing methods either provide fixed, domain-specific mock tools (Styles et al., 2024; Liu et al., 2024b; Chen et al., 2025a) or simply wrap real APIs (Qin et al., 2024), which has limited general-purpose tool simulation. For example, Tool-Sandbox (Lu et al., 2025) and BFCLv3 (Patil et al., 2025) provide a set of stateful tools whose outputs depend on history tool executions to simulate multi-turn tasks. $\tau$-bench (Yao et al., 2025) and $\tau^2$-bench (Barres et al., 2025) emulate conversations between a user and an agent in airline and retail scenarios. While these methods provide precise simulation on the designed use cases, they are limited by human-written tools and datasets and are hard to generalize. Therefore, they are primarily used for evaluation rather than to improve LLM performance at test time.

## 3. The Gecko Simulation Environment

Gecko has five components: (1) an argument validator that checks the syntactic and semantic validity of tool calls (Section 3.1); (2) a response generator that synthesizes realistic

| Detection rate | GPT-4o | GPT-4.1-nano | Qwen-2.5-7B |
|---|---|---|---|
| True positive | 100% | 100% | 100% |
| Syntactic errors | 100% | 100% | 100% |
| Semantic errors | 71% | 69% | 63% |

*Table 1.* Accuracy of argument validation on BFCL-Live-Simple. 'True positive' is the rate where correct arguments are determined as correct. We also show percentage of syntactic errors and semantic errors being detected. As helper LLMs, GPT-4o, GPT-4.1-nano and Qwen-2.5-7B are evaluated.

outputs for validated tool calls (Section 3.2); (3) a task state estimator that keeps track of the evolving task state (Section 3.3); (4) a task feedback generator that judges task completion and identifies remaining objectives (Section 3.4); and (5) an API schema converter that transforms new tools into OpenAPI 3.1 schemas for integration (Section 3.5).

### 3.1. Argument Validator

**Syntactic validation with schema-based rules**. Before simulating tool responses, Gecko first checks whether a tool call is executable under the corresponding tool definition. The rule-based validator verifies that the called tool exists, all required arguments are present, unsupported arguments are rejected, and basic data types such as integer, string, and boolean are satisfied. It also enforces schema-expressible constraints, including predefined ranges, enum values, and format requirements when such constraints are explicitly specified. Additional rule-based checks are listed in Appendix C. Violations of these checks result in *error feedback*; examples are shown in Fig. 3(a).

**Semantic validation with a helper LLM**. Some argument constraints are not formally encoded in the tool schema, but are instead described in natural-language tool descriptions or implied by the task context. Gecko therefore uses a helper LLM as a complementary validator for such description-level and context-dependent constraints, rather than as a replacement for schema-based rule checks. Given the tool schema, user task, and candidate arguments, the helper LLM checks whether each argument value is semantically consistent with the expected use of the tool. For example, consider a tool `get_stock_price(symbol: string)` whose description states that `symbol` should be a stock ticker. The argument `symbol = Apple Inc.` satisfies the primitive string type, but violates the description-level requirement because the tool expects a ticker such as `AAPL` rather than a company name. When such semantic inconsistencies are detected, Gecko returns *error feedback*. Examples are shown in Fig. 3(b).

Table 1 presents the accuracy of argument validation on BFCL-Live-Simple. We use three metrics: true positive detection rate (correct arguments detected as correct), syntactic error detection rate, and semantic error detection rate.

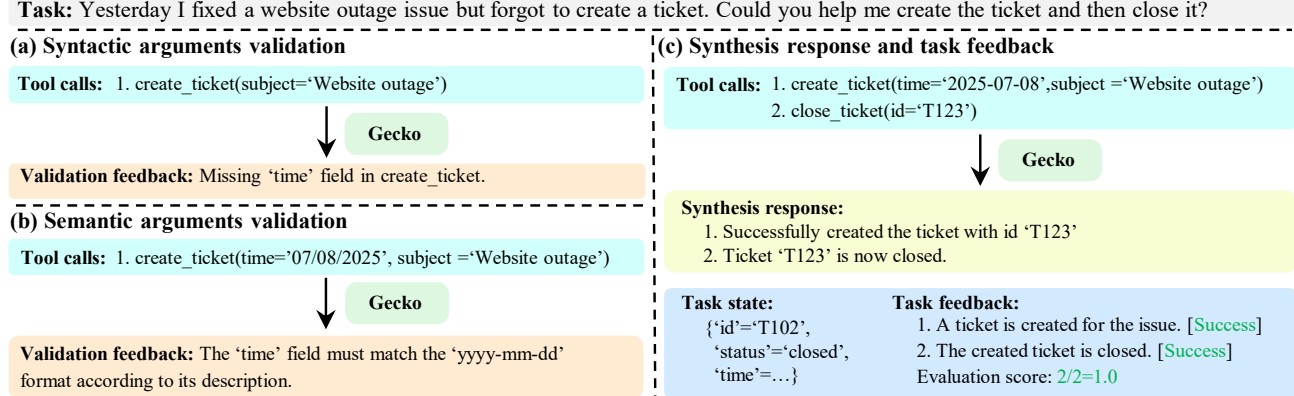

*Figure 3.* Examples of feedback provided by Gecko through (a) syntactic argument validation, (b) semantic argument validation, and (c) synthetic responses, task state, and task feedback. The validation feedback (orange), synthetic response (yellow), and task state and feedback (blue) will then be fed to the planning LLM during test-time scaling.

Details of how the three metrics are computed are provided in Appendix C. Gecko has 100% accuracy in finding correct arguments and detecting syntactic errors for all three helper LLMs. When there are semantic errors in the tool arguments, Gecko can detect $60\% \sim 70\%$ of them, showing that description-level validation can filter out a substantial fraction of invalid calls but remains imperfect. As shown in Fig. 4(a), accuracy drops by 1.5% if argument validation is removed, demonstrating its usefulness.

### 3.2. LLM-based Tool Response Generator

Beyond using validated arguments and the tool schema as inputs, the response generator synthesizes a schema-conforming output that approximates the behaviour of the corresponding real tool. To support multi-turn tool-use tasks, Gecko additionally conditions response generation on the current task state (Section 3.3), a compact representation of prior tool-call effects. This helps reduce factual conflicts with earlier tool responses and preserves cross-call consistency. The generated responses are then used by the task state estimator and serve as an important source of feedback for subsequent refinement. Examples are shown in Fig. 2 and Fig. 3(c), and prompts are provided in Appendix D.

It is non-trivial to directly measure the effectiveness of response generation because there is no ground truth. We therefore conduct an indirect task-level evaluation on BFCL-Multi-Turn-Base to measure whether simulated responses provide useful evidence for downstream task-success judgment. Specifically, we sample GPT-4o solutions and use the official BFCL evaluator to label each solution as successful or failed. We then ask Gecko's task feedback generator to judge the same solutions under two settings: one conditioned on real-tool executions and the other conditioned on simulated-tool executions. This comparison evaluates both the reliability of the judge under real executions and the ad-

|                            | True positive | True negative |
|----------------------------|:-------------:|:-------------:|
| Real tools, LLM judge      |     96.7%     |     87.5%     |
| Simulated tools, LLM judge |     91.3%     |     87.5%     |

*Table 2.* True positive and true negative rates of the LLM judge for task success/failure judgments. We compare judgments based on trajectories generated with real tools and simulated tools.

ditional degradation introduced by replacing real tools with simulated responses. As shown in Table 2, under real-tool executions, the judge achieves 96.7% true positive rate and 87.5% true negative rate. Replacing real tools with simulated tools reduces the true positive rate to 91.3%, while the true negative rate remains 87.5%. This suggests that simulated responses are imperfect but still provide informative task-level evidence for pre-execution refinement.

### 3.3. LLM-based Task State Estimator

The task state is a compact textual representation of task progress in Gecko. It summarizes the cumulative effects of previous tool calls and serves as the shared context for later response generation and task-level judgement. Given the previous task state, the tool call, and the simulated response, a helper LLM updates the state to reflect the effect of that call. For example, given the previous state "apples in cart: 3" and a tool response "successfully removed one apple," the updated state would be "apples in cart: 2." The task state is used by the response generator (Section 3.2) to preserve cross-call consistency and by the task feedback generator (Section 3.4) to judge whether the user objective has been satisfied. For more examples, see Fig. 3(c) and Fig. 2.

Similar to task response generation, there is no direct measurement of task state estimation performance. We therefore rely on the downstream task-success judgement in Table 2 as an indirect evaluation. The strong true positive and true neg-

ative rates suggest that the estimated states provide useful information for determining task completion.

### 3.4. LLM-based Task Feedback Generator

We use a judge LLM to create feedback to be sent to the planning LLM. There are two steps. First, we use the task description as input and let the judge LLM generate a checklist specifying what aspects are important for indicating the completion of this task, such as 'The *temp* folder is created in *document*' and 'The *report.pdf* is moved to *temp*' for the task in Fig. 2. Second, we let the judge LLM decide whether the Gecko execution results fully satisfy the checklist: if yes, then the task feedback indicates success; if not fully satisfied, the judge LLM identifies remaining objectives, and another round of LLM planning and Gecko simulation will be executed. Specifically, the input to the judge LLM includes the task, tool calls from the planning LLM, the simulated responses (Section 3.2), and the task state (Section 3.3). An example of task feedback is shown in Fig. 3(c).

In Table 2, we present the accuracy of task success/failure judgement on the BFCL-Multi-Turn-Base. When using real tools, the LLM judge achieves a true positive rate of 96.7% and a true negative rate of 87.5%. This indicates that our task feedback generator works well.

The components described from Section 3.1 to Section 3.4 allow us to finally ground the agent tool calls. That is, after receiving the tool calls from the planning LLM, Gecko checks argument validity, simulates tool responses, estimates task state, and then gives task feedback.

### 3.5. LLM-based Schema Converter for Various Tools

Tools with standard OpenAPI schemas can be directly integrated into Gecko. For Python functions and other non-OpenAPI tool definitions, Gecko uses an LLM-based schema converter to produce OpenAPI schemas. Given a description of a tool that details its purpose, input parameters, and expected output, an LLM generates an OpenAPI 3.1.0 specification. A rule-based OpenAPI schema validator is used to ensure the generated schema is syntactically correct. Our text prompt is provided in Appendix G. This conversion step allows Gecko to integrate tools from different definition formats into a unified schema interface. The resulting schemas are used by the argument validator (Section 3.1) to check tool calls and by the response generator (Section 3.2) to synthesize schema-conforming outputs.

## 4. Grounding Agent Test-Time Scaling (GATS)

GATS is an inference-time refinement method that uses Gecko as a simulated execution environment before the real-tool execution. Given a user task, GATS first lets the planning LLM generate a candidate tool-call trajectory in

Gecko. For each simulated attempt, Gecko behaves like a tool execution environment from the planning LLM's perspective: each tool call either returns an error message from argument validation or a simulated tool response. If a tool call is invalid, Gecko immediately returns validation feedback. If the call is valid, Gecko synthesizes a tool response and updates the task state according to the task-relevant effect of the call.

After the planning LLM finishes a candidate trajectory, Gecko's task feedback generator evaluates the latest task state and the full tool-call sequence against the user task. It then judges the attempt as either *task completed* or *task failed*. If the attempt fails, Gecko returns the failed trajectory and task-level feedback to the planning LLM, which uses them to refine the next simulated attempt. To prevent failed attempts from modifying the environment for later retries, Gecko evaluates each retry in an isolated simulation session initialized from the same state snapshot. This session-based isolation mechanism, together with Gecko's task-state recording, is described in Appendix B.

GATS repeats this process until Gecko judges a simulated trajectory as completed or the retry budget is exhausted. When a completed trajectory is found, GATS prepends its tool-call plan to the original user message as in-context guidance for the final real-tool agent. The final agent then solves the task using real tools, guided by the successful simulated plan rather than by directly executing the simulated outputs. If no simulated attempt succeeds, no guidance is added and the final agent reduces to the original baseline setting. A diagram of this iterative scaling method is shown in Fig. 2.

## 5. Discussion

**Can Gecko be implemented only by prompting?** Technically yes if we can find a *perfect* prompt to let the LLM output all feedback and responses. But such a prompt is almost impossible to find, because 1) our system is a combination of rules and LLM use, and 2) even if the prompt is successfully written, it will be too complex for LLMs to understand. That said, we implemented a merge-to-one method, which collapses Gecko into a single prompt; it performs substantially worse than GATS (see Table 5), suggesting that prompt-only emulation is insufficient.

**Do simulation errors accumulate?** Yes and no. For single-turn tasks, errors will accumulate. For multi-turn tasks, real tools are used at the end of each turn, so the correct task state will be obtained. It means accumulation is bounded. That said, regardless of error accumulation, a single error in tool call will mean task failure according to the current evaluation protocol. In the real-world multi-turn scenarios, errors can be corrected by users through communication.

**Comparison with StableToolBench (STB) (Guo et al.,**

*Table 3.* Method comparison on BFCLv3. We select eight most important metrics from BFCL website. Overall accuracy is the average of average 'Non-live single turn', average 'Live single turn', and 'Multi-turn'. GATS consistently improves various planning LLMs.

| Model | Overall Acc | Non-live single turn | | | | Live single turn | | | Multi-turn |
|---|---|---|---|---|---|---|---|---|---|
| | | simple | parallel | multiple | irrelevance | simple | multiple | irrelevance | base |
| **State-of-the-art reference models** | | | | | | | | | |
| ToolACE-2-8B | 73.12 | 88.00 | 92.50 | 92.50 | 95.41 | 70.93 | 79.01 | 84.80 | 49.00 |
| watt-tool-70B | 79.27 | 98.25 | 85.50 | 94.00 | 84.16 | 86.04 | 83.47 | 68.48 | 68.00 |
| xLAM-2-70b | 80.96 | 94.75 | 92.00 | 94.50 | 83.33 | 77.13 | 71.13 | 74.48 | 77.50 |
| **Baseline models and our proposed method** | | | | | | | | | |
| GPT-4.1-nano | 58.85 | 82.25 | 78.50 | 75.00 | 80.83 | 65.11 | 58.97 | 72.22 | 32.00 |
| +GATS | 67.59 | 93.25 | 88.50 | 95.00 | 81.25 | 77.13 | 69.80 | 80.38 | 37.50 |
| GPT-4.1-mini | 66.20 | 91.50 | 84.50 | 88.00 | 78.33 | 79.45 | 70.94 | 68.70 | 40.00 |
| +GATS | 73.84 | 96.25 | 88.00 | 95.50 | 84.58 | 84.49 | 74.54 | 80.83 | 50.50 |
| GPT-4o | 76.93 | 92.75 | 92.50 | 92.50 | 84.16 | 81.00 | 78.53 | 78.45 | 61.00 |
| +GATS | 84.62 | 96.50 | 95.00 | 95.50 | 95.83 | 84.10 | 81.01 | 93.42 | 72.00 |
| GPT-5 | 61.94 | 78.00 | 84.00 | 76.00 | 92.91 | 61.62 | 57.45 | 89.70 | 33.50 |
| +GATS | 66.08 | 85.00 | 90.50 | 83.00 | 93.75 | 67.44 | 63.24 | 90.38 | 36.50 |
| Gemini-2.5-pro | 66.44 | 86.25 | 69.00 | 86.00 | 91.66 | 77.90 | 62.20 | 89.68 | 39.50 |
| +GATS | 70.44 | 92.25 | 75.00 | 89.00 | 92.50 | 80.62 | 67.99 | 91.83 | 44.00 |
| Gemini-3.0-pro | 79.97 | 94.50 | 91.00 | 94.00 | 82.50 | 87.60 | 80.44 | 73.19 | 69.00 |
| +GATS | 85.19 | 97.00 | 93.00 | 95.50 | 94.17 | 85.93 | 82.34 | 89.59 | 73.50 |
| Deepseek-V3 | 70.40 | 97.00 | 92.00 | 94.00 | 80.41 | 86.04 | 79.48 | 72.56 | 41.00 |
| +GATS | 72.90 | 97.25 | 92.00 | 95.50 | 83.75 | 88.75 | 81.76 | 78.79 | 43.50 |
| Qwen-3-14B | 73.78 | 95.50 | 92.50 | 95.00 | 84.58 | 86.04 | 80.81 | 77.44 | 48.00 |
| +GATS | 78.60 | 96.75 | 93.50 | 95.00 | 92.50 | 87.59 | 83.00 | 91.50 | 54.00 |
| GPT-5.5 | 75.63 | 91.50 | 90.50 | 88.50 | 88.75 | 73.64 | 73.79 | 83.82 | 60.00 |
| +GATS | 81.11 | 94.00 | 91.00 | 96.50 | 89.17 | 88.37 | 80.15 | 86.99 | 65.50 |

**2024**). While both Gecko and STB can simulate API responses, Gecko has a few key advantages. **First**, to simulate an API, STB needs to collect real responses from this API, which can be costly and less flexible. In comparison, Gecko directly supports new APIs using only API descriptions. **Second**, STB lacks API argument validations and generates responses for all API calls, including invalid ones, whereas real-world API servers reject such invalid calls. In contrast, Gecko has an argument validity checker that rejects invalid API calls and returns meaningful error messages, thus more closely following common API-server behavior. **Third**, STB focuses on individual API calls without considering multi-turn conversation history, while Gecko considers history from both task states and conversation, especially in multi-turn scenarios, which ensures logical coherence.

**New research possibilities enabled by Gecko. First**, Gecko is complementary to existing tool-call data synthesis pipelines (Liu et al., 2024c;a) as a verifier to improve dataset quality. A typical tool-call data point contains a task, tool definitions, and a tool-call sequence. These could be fed into Gecko, which would simulate tool responses, estimate task state, and return task feedback indicating whether the tool-call sequence solves the task. This feedback can be used to filter out or correct erroneous data points. **Second**,

Gecko can turn existing supervised fine-tuning (SFT) tool-call datasets into reinforcement learning (RL) environments. Given tool definitions, Gecko can form an action space by converting each tool definition to a callable tool. After each action, Gecko returns an observation that simulates the tool execution result. Reward signals can be derived from an LLM judge via checklist–state comparison. The resulting trajectories can be used for offline RL, and the same interface supports online exploration in Gecko.

**Limitations.** Gecko should not be viewed as a high-fidelity replacement for arbitrary real tools. It is most suitable for tool-use settings where tool specifications and task context provide enough information to generate useful simulated feedback. It is less suitable for tools whose behavior depends on hidden specialized knowledge, exact runtime states, rapidly changing backends, or non-text outputs. For example, Gecko currently supports text-output tools, such as `get_temperature`, but does not yet support tools whose outputs are non-text media, such as `download_video`. In addition, for tools that rely on hidden external databases or changing backend states, such as airline reservation systems, simulated outputs (*e.g.*, available flights or user bookings) may differ from real execution results. For real-world deployment of GATS, one mitigation strategy is a hybrid

mode: Gecko simulates state-changing write tools while directly executing read-only or query tools. This design reduces simulation-reality drift for information retrieval while preserving Gecko's sandbox benefits for pre-execution refinement. More broadly, Gecko is intended to provide informative refinement signals before real execution, rather than to fully substitute for real tools.

# 6. Experiments

## 6.1. Experimental Setup

**Benchmark.** We evaluate Gecko and the GATS method on the Berkeley Function Call Leaderboard v3 (BFCLv3) and the $\tau^2$-bench. **BFCLv3** evaluates LLM tool-use ability in three categories: non-live single-turn, live single-turn, and multi-turn. Single-turn means that a task must be completed in one user–assistant round; non-live indicates that tasks and tools are designed by experts, while live indicates that tasks and tools are sourced from real-world scenarios. Multi-turn requires the model to plan and generate tool calls across several rounds based on tool-execution results and the user feedback. Within single-turn, there are four task types: simple (one tool call is executed to answer a user query), multiple (multiple tool calls are executed sequentially to answer a user query), parallel (multiple tool calls are executed in parallel to answer a user query), and irrelevance (none of the provided tools is appropriate, so the correct behavior is to avoid tool use). In total, BFCLv3 contains 3,633 tasks involving 8,578 tools. $\tau^2$-**bench** is specifically designed to assess agent abilities in real-world retail scenario ($\tau^2$-retail) and airline scenario ($\tau^2$-airline). In $\tau^2$-bench, the agent should communicate with an LLM-simulated user, call domain APIs and follow domain policy rules (*e.g.*, refund and booking rules) to complete tasks. $\tau^2$-retail has 13 APIs and 114 tasks; $\tau^2$-airline has 12 APIs and 50 tasks.

**Evaluation metrics.** For BFCLv3, we report accuracy, defined as the percentage of tasks completed correctly. For **single-turn** tasks, a prediction is counted as correct only if the tool calls produced by planning LLM exactly match the reference solution. For **multi-turn**, correctness is judged by comparing task outcomes after each turn, such as tool results and updated file content, with the ground truth. A multi-turn task is considered successful only if task outcomes match the ground truth at every turn. For $\tau^2$-bench, we use pass@1 averaged over 4 independent runs per task. Each task has an annotated goal database state, and a run is successful only if the agent provides all required information and the final database matches the annotation.

**Implementation details.** For each planning LLM, we use the same backend model as the helper/judge LLM for tool response generation, task state estimation, and task feedback generation, because these components require stronger se-

*Table 4.* Method comparison on $\tau^2$-bench. We report success rate under $\tau$-retail and $\tau$-airline subsets and average accuracy (Overall).

| Model | $\tau^2$-retail | $\tau^2$-airline | Overall |
|---|---|---|---|
| **State-of-the-art reference models** | | | |
| Claude Opus 4 | 81.8% | 60.0% | 70.9% |
| Claude Sonnet 4 | 75.0% | 55.5% | 65.3% |
| Kimi-K2-Instruct | 70.6% | 56.5% | 63.6% |
| **Baseline models and w/ GATS** | | | |
| GPT-4o | 62.9% | 45.5% | 54.2% |
| +GATS | 69.3% | 52.0% | 60.7% |
| GPT-5-mini | 73.5% | 57.0% | 65.3% |
| +GATS | 78.5% | 65.0% | 71.8% |
| GPT-5-thinking | 81.6% | 63.0% | 72.3% |
| +GATS | 84.6% | 68.0% | 76.3% |
| Gemini-3.0-pro | 85.1% | 72.5% | 78.8% |
| +GATS | 88.2% | 76.5% | 82.3% |
| GPT-5.5 | 85.1% | 86.0% | 85.5% |
| +GATS | 87.7% | 88.0% | 87.9% |

mantic reasoning. This design avoids introducing a stronger model as an additional confound and attributes performance changes more directly to Gecko and GATS. For BFCLv3, we run GATS with a maximum retry budget of 3. When Gecko judges a simulated trajectory as successful, the trajectory is provided as in-context guidance to the real-tool agent, which then solves the task in the benchmark environment. If no simulated attempt succeeds, no guidance is added and the method reduces to the original baseline setting.

$\tau^2$-bench differs from BFCLv3 because its internal databases, such as prior reservations and available flights, are not exposed to the agent. If Gecko simulates read-only database-query tools, it must also simulate the hidden database state, which can introduce substantial simulation-reality drift. To mitigate this issue, during GATS refinement on $\tau^2$-bench, we execute read-only tools such as get_user_details against the benchmark-provided real databases, while keeping state-changing write tools simulated in Gecko. This hybrid setting does not remove Gecko's main sandbox benefit, because read-only tools do not modify external state, while potentially harmful or irreversible state-changing actions are still explored in simulation. In $\tau^2$-retail, 6 out of 13 tools are read-only; in $\tau^2$-airline, 6 out of 12 tools are read-only.

## 6.2. Main Evaluation

**GATS consistently improves tool use capabilities of existing LLMs.** On BFCLv3 and $\tau^2$-bench, we use various existing LLMs as the planning LLM, such as GPT-4.1-mini (OpenAI, 2025a), DeepSeek V3 (DeepSeek-AI et al., 2025), watt-tool-70B (watt-ai, 2025), Qwen3-14B (Yang

*Table 5.* Comparison between GATS and existing test-time scaling methods. We report performance, average token usage, and average number of real benchmark tool calls per task. Simulated tool calls inside Gecko are excluded from the tool-call count. GATS achieves the best performance in both settings while keeping real-tool overhead reasonable.

**(a) $\tau^2$-airline with GPT-5-mini**

| Method | Perf. | Tokens (k) | Avg. T.C. |
|---|---|---|---|
| GPT-5-mini | 57.0% | 80.3 | 6.70 |
| w/ Reflexion | 60.0% | 102.9 | 6.62 |
| w/ Merge-to-one | 53.5% | 111.2 | 5.88 |
| w/ Self-Refine | 59.0% | 259.6 | 23.88 |
| w/ Best-of-N | 58.5% | 365.8 | 13.78 |
| w/ Majority Voting | 59.0% | 291.7 | 6.62 |
| w/ GATS | **65.0%** | 238.2 | 7.76 |

**(b) BFCLv3 Multi-Turn-Base with GPT-4o**

| Method | Acc. | Tokens (k) | Avg. T.C. |
|---|---|---|---|
| GPT-4o | 61.0% | 30.2 | 7.33 |
| w/ Reflexion | 63.5% | 36.1 | 8.10 |
| w/ Merge-to-one | 58.0% | 61.6 | 7.46 |
| w/ Self-Refine | 64.5% | 76.9 | 16.22 |
| w/ Best-of-N | 63.0% | 78.7 | 14.04 |
| w/ Majority Voting | 66.0% | 129.3 | 8.67 |
| w/ GATS | **72.0%** | 108.7 | 6.69 |

et al., 2025), Kimi-K2-Instruct (Kimi Team et al., 2025), Claude Opus 4 (Anthropic, 2025), GPT-5 (OpenAI, 2025b), GPT-5.5 (OpenAI, 2026) and Gemini-3.0-pro (Google DeepMind, 2026). We summarize the performance in Table 3 and Table 4 and have four observations.

**First**, GATS consistently improves tool-call performance of these LLMs. For example, on BFCLv3, the overall accuracy of GPT-4o and Qwen-3-14B is improved from 76.93% and 73.78% to 84.62% and 78.60%, respectively. On $\tau^2$-bench, our method improves GPT-4o from 54.2% to 60.7%, and GPT-5 from 72.3% to 76.3%. **Second**, our method is effective for both single-turn and multi-turn tasks. For example, the improvement of GPT-4o on 'Live single turn' is +3.10% and +2.48% for 'simple' and 'multiple', respectively. **Third**, GATS is useful for both retail and airline scenarios, where we generally observe 3%-8% pass@1 gain. **Fourth**, while some planning LLMs have different performance on single-turn tasks, GATS may bring them to similar levels. For example, on 'Multiple' under 'Non-live single turn', the performance of GPT-4.1-mini, GPT-4.1-nano, DeepSeek-V3, and GPT-4o becomes ~95% from 88%, 75%, 94%, and 92.5%, respectively. It suggests there exists some upper limit of Gecko or BFCLv3 itself (*e.g.*, annotation errors).

**Comparison with the state of the art.** We apply GATS to Gemini-3.0-pro and report new state of the art (SOTA) on BFCL: **overall accuracy = 85.19%**. Moreover, on various subsets, Gemini-3.0-pro+GATS also reports very competitive performance, *e.g.*, 97.00% on simple single turn and 95.50% on multiple single turn. The multi-turn-base performance, 73.50%, is the second best among all the methods. For $\tau^2$-bench, Gemini-3.0-pro with GATS reports **an overall accuracy of 82.3%**, which is very competitive performance compared with SOTA.

### 6.3. Further Analysis

**Comparison with existing test-time scaling methods.** In Table 5, we compare GATS with existing inference-time refinement methods, including Reflexion (Shinn et al., 2023), Self-Refine (Madaan et al., 2023), Best-of-N, Majority Vot-

ing, and a Merge-to-one variant of Gecko. For Best-of-N, we sample $N = 3$ candidates with temperature 0.7. For Majority Voting, we sample $N = 5$ candidates with temperature 0.7. Merge-to-one collapses Gecko's components into a single prompt, testing whether prompt-only emulation can replace Gecko's structured simulation environment. We report results on two settings: $\tau^2$-airline with GPT-5-mini, and BFCLv3 Multi-Turn-Base with GPT-4o. For tool-call overhead, we count only real benchmark tool calls.

On $\tau^2$-airline, GATS achieves the best pass@1 rate of 65.0%, outperforming the second-best method, Reflexion, by 5.0 percentage points. On BFCLv3 Multi-Turn-Base, GATS again achieves the best accuracy, improving GPT-4o from 61.0% to 72.0% and outperforming Majority Voting by 6.0 percentage points. GATS uses more tokens than lightweight baselines such as Reflexion and Merge-to-one, but fewer tokens than Best-of-N on $\tau^2$-airline and fewer tokens than Majority Voting on BFCLv3 Multi-Turn-Base. In terms of real tool calls, GATS keeps the overhead reasonable: it uses far fewer real tool calls than Self-Refine and Best-of-N in both settings. Overall, these results show that Gecko-based simulated refinement provides strong accuracy gains while avoiding excessive real-tool overhead.

**Ablation studies.** Gecko has four key components: argument validation, task state estimator, response generator, and feedback generator. Among the four, response generation cannot be removed, so our ablation studies are for the remaining three. We experiment on the BFCL-Multi-Turn-Base with GPT-4o. Results are shown in Fig. 4 (a). *w/o arg validation* removes both rule-based and LLM-based argument validation in Gecko. *w/o task state est.* does not estimate the task states. *w/o task feedback* replaces the judge LLM with a naive gating mechanism: if tool calls are generated, we give a success feedback; if no tool calls are generated, then failure feedback. Results show that removing argument validation slightly decreases accuracy (from 72.0% to 70.5%), while removing task state estimation has a greater impact (68.0%). Eliminating task feedback causes the most significant drop (61.5%), indicating that the feedback is most important when solving multi-turn tasks.

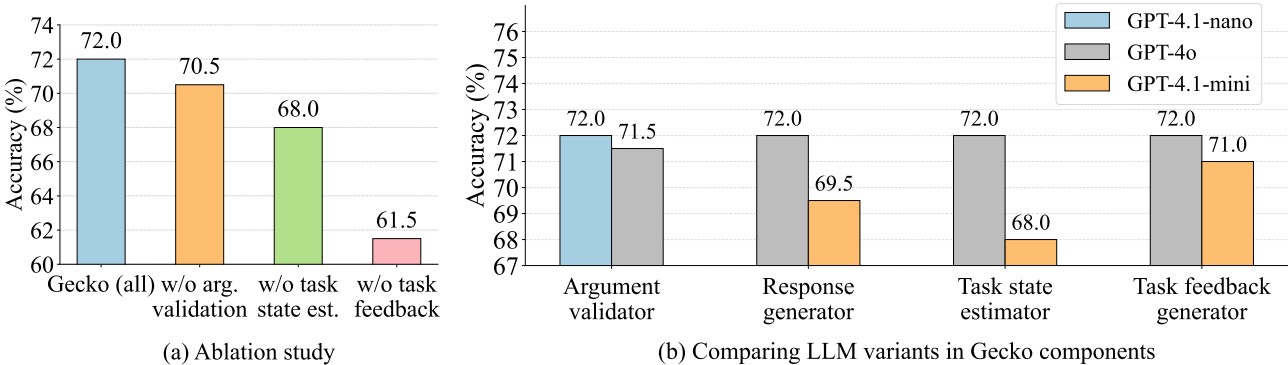

(a) Ablation study          (b) Comparing LLM variants in Gecko components

*Figure 4.* (a) Ablation study of Gecko on BFCL-Multi-Turn-Base. The full system (72.0%) is compared against variants with one component removed: argument validation (70.5%), task state estimation (68.0%), and task feedback (61.5%); (b) LLM replacement study on Gecko evaluated on BFCL 'Multi-turn base'. We use GPT-4o as planning LLM. For each component, the bar on the left is the original performance 72.0%. Under LLM replacement, *e.g.*, replacing GPT-4.1-nano with GPT-4o for argument validation.

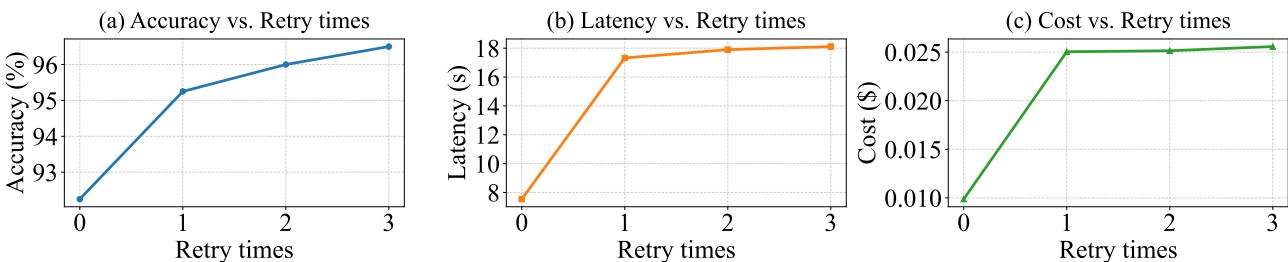

*Figure 5.* Test-time scaling behaviours. We evaluate GATS on the BFCL-Non-Live-Simple, using GPT-4o as the planning LLM. *Retry times* is the maximum number of feedback-based refinement steps allowed in GATS (Section 4). We report (a) accuracy (%), (b) average latency (s) and (c) average cost ($) per user task versus maximum retry times.

**Comparing different LLMs used in different components in Gecko.** To investigate the impact of using different helper/judge LLMs, we replace them with a different LLM while keeping the other components unchanged. Results on BFCL-Multi-Turn-Base are shown in Fig. 4(b), where the performance of the default setting is 72.0%. Replacing GPT-4.1-nano in argument validation with GPT-4o results in a very minor drop (71.5%). Because argument validation is relatively simple, it does not require strong LLMs like GPT-4o. If we replace GPT-4o with GPT-4.1-mini in response generation and task state estimation, performance drops from 72.0% to 69.5% and 68.0%, respectively. For the task feedback components, replacing GPT-4o with GPT-4.1-mini leads to a small decrease in performance (-1.0%). This shows the robustness of this component to weaker LLMs.

**Scaling behavior of GATS.** We examine how the retry budget affects performance on the BFCL-Non-Live-Simple, where maximum retry times vary from 0 to 3. As shown in Fig. 5, increasing the max retry times improves accuracy, from 92.25% with no refinement to 96.50% with three times of refinement. Most accuracy gain comes from the first retry, while further retries add less improvement. Accordingly, latency and cost increase with retry times: runtime increases

from 7.54 s to 18.11 s, and cost from $0.00987 to $0.02557. Both also demonstrate decreasing margin: most user tasks are resolved in the first retry, so they will not use up the maximum retry times. These results clearly demonstrate a trade-off between accuracy gain and cost.

## 7. Conclusion

This paper introduced Gecko, a stateful simulation environment for pre-execution refinement of LLM agent tool calls. Given tool calls from a planning LLM, Gecko provides validation feedback, simulated tool responses, and task-level feedback. The feedback allows us to propose GATS, a test-time scaling method that iteratively refines tool calls in simulation before real-tool execution. This method consistently improves various LLMs during test time on agentic tool use benchmarks and reports state-of-the-art performance. In the future, Gecko can serve as foundational infrastructure for agentic tool use, enabling the community to (1) improve agent tool use at test time via tool simulation and informative feedback; (2) synthesize high-quality tool-call data for supervised fine-tuning (SFT); and (3) turn SFT tool-call datasets into reinforcement learning environments.

## Impact Statement

The goal of our work is to advance the field of large language models. There are many potential societal consequences of our work, none of which we feel must be specifically highlighted here.

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

## A. The Use of Large Language Models (LLMs)

OpenAI ChatGPT was used for grammar checks and phrasing suggestions. Anthropic Claude (Claude Code) was used to assist with implementing and debugging portions of the code. LLMs did not contribute to research ideation or experimental design.

## B. Design Principles of Gecko

Gecko is a simulated tool execution environment built on modern web-service principles and accessible over the network. It exposes a RESTful HTTP interface so clients can interact with it using standard JSON payloads.

Gecko uses **FastAPI**, a high-performance asynchronous web framework, to handle API requests, and **CAMEL** (Li et al., 2023) as the LLM agent framework. FastAPI's native async support and lightweight routing let Gecko handle many concurrent requests with low latency.

The architecture follows a **middleware-based design pattern** that provides a clear separation of responsibilities. Each incoming request passes through a chain of middleware components that handle different aspects of the processing pipeline. The session middleware manages state isolation between different execution contexts, while the route middleware handles the mapping between API endpoints and their corresponding OpenAPI specifications. This layered approach ensures that each component focuses on a specific responsibility while maintaining loose coupling between different parts of the system.

To support concurrency and retry mechanisms, we designed a **session-based state management system**. Each client interaction runs inside its own session with a unique session ID. The session holds its own configuration, execution history, and task state history so the server can safely retry actions and replay earlier steps without mixing data from different clients. This isolation prevents users from interfering with each other and lets Gecko handle many users at once. Sessions persist across multiple tool calls, enabling multi-turn interactions that remain tied to the same session.

## C. Argument Validator Implementation Details

### C.1. Validation Flow

Each tool call follows a fixed validation pipeline before execution. The server first parses the tool call, then routes it to the corresponding OpenAPI operation by a FastAPI layer. The schema related to the OpenAPI operation is the single source of truth for validation. We then apply (1) deterministic, rule-based checks from the schema and (2) an optional LLM-based semantic check that inspects schema descriptions.

### C.2. Syntactic Validation Rules

Our syntactic validation enforces comprehensive rule-based checks to ensure argument correctness at the structural level.

**Presence of required fields.** We verify that all fields marked `required` in the OpenAPI schema are present in the tool call. We also check that the tool call does not include any fields that are not defined in the schema. Tool calls that violate this rule will receive an instant error message.

**Type checking.** Each provided argument is checked against the type (e.g., `integer`, `float number`, `string`, `boolean`, `array`, and `object`) declared in the OpenAPI schema. Type mismatches will result in an error message.

**Constraint enforcement.** The validator enforces constraints such as numeric bounds (`minimum`/`maximum`), enumeration (`enum`), and length/pattern restrictions for strings. When constraints are violated, the validator reports the specific constraint failure.

### C.3. Semantic Validation via LLM

The semantic validator takes two primary inputs: the tool call to be verified and the corresponding OpenAPI schema. The validator returns a JSON-parsable message that indicates whether the tool call is semantically acceptable and enumerates any problems found. The main prompt for the semantic validator is provided below.

Please validate the given function call arguments against their parameter schemas.

**Validation Rules**:
1. **Scope**
   – Only validate arguments defined in the provided schemas.
   – Ignore arguments not present in the schema (do not treat them as errors).
   – Type validation has already been handled elsewhere. Just skip type checking.

2. **Semantic Checks**
   – Validate according to the parameter description, examples, enums, or format requirements.
   – If examples are provided (e.g. "full−time, part−time"), treat them as semantic categories. Any value in the same
   ↪ category (e.g. "internship", "contract") is valid.
   – If the description specifies a format (e.g. 'YYYY−MM−DD'), enforce that exact pattern.
   – Use common sense to ensure values are within a reasonable range (e.g. interest rate within [0,1]; clock hour
   ↪ within [0,12]).
   – Detect redundant/overlapping information across arguments (e.g. 'item="large pizza"' and 'size="large"' are
   ↪ overlap).
   – If uncertain about validity, default to considering the argument valid.

3. **Error Messages**
   – Concise, precise, and human−readable.
   – Do not include or suggest correct values.
   – Only state which argument is invalid and why.

**Output Format**:
'''

valid=\<true|false> error\_message="\<if false, list each invalid argument and reason>"

'''

**Example**:
− **params_schema**
  '[{"location": "The city that you want to go, e.g. 'Beijing, China'"}, {"date": "The start date for the booking,
  ↪ format: YYYY−MM−DD"}]'
− **args**
  '{"location": "London", "date": "01/01/2024"}'
− **Output**
  'valid=false error_message="location not in required format (should include city and country); date not in required
  ↪ format (YYYY−MM−DD)"'

# D. Response Generator Implementation Details

The response generator synthesizes a tool response given three inputs: a tool call, the corresponding OpenAPI schema, and the current estimated task state. The main prompt for the response generator is provided below.

---

You are an API simulation engine that generates JSON responses strictly following OpenAPI 3.1 schemas.

Rules:
1. Schema first. Always match the schema exactly (structure, names, types, formats, required fields).
2. Entity−level consistency. Do not contradict any provided state or any prior successful responses in this session.
3. Open−world reads. For read/query/search operations, if requested entities/data are absent in the provided state,
↪ you MUST synthesize realistic, schema−compliant values instead of returning not−found or error responses.
4. Writes remain consistent. For create/modify/delete operations, produce a success result consistent with the
↪ schema unless it would contradict previously returned state; do not invent conflicts.
5. No extra rules. Do not invent constraints beyond the tool definition and the provided state.

Realism & uniqueness guidelines (domain−agnostic):
− Deterministic diversity: derive identifier−like fields using stable transforms of input arguments (e.g., incorporating
↪ parts of arguments or their hashes) so that different arguments yield different values within the session, while the
↪ same arguments yield stable values.
− Identifier−like fields (e.g., keys ending with '_id', 'Id', 'code', 'number'): prefer distinct values for distinct entities
↪ in the same response unless the schema indicates they refer to the same entity.
− Consistency: when two items share the same identifier−like value in one response, their associated attributes
↪ MUST NOT contradict each other within that response.
− Plausible formats: choose values that look realistic when the schema allows free−form text, but always prioritize
↪ matching schema types and formats.
− Temporal consistency when both present: end/arrival timestamps should be after start/departure timestamps;
↪ choose plausible intervals without assuming domain−specific constraints.
− Diversity: vary counts and enumerations when optional, within reasonable ranges, while staying
↪ schema−compliant.

Illustrative synthesis examples:
− Tool name: get_user_details:
  Request: {"user_id": "john_doe_001"}
  Response (success object):
  {
    "user_id": "john_doe_001",
    "name": "John Doe",
    "email": "john.doe@example.com",
    "phone": "+1−222−345−6789",
    "loyalty_status": "silver",
    "miles": 50000,
    "address": "7340 Oak Street, San Francisco, CA 94110"
  }

---

# E. Task State Estimation Implementation Details

The task state estimation contains two phases: initialization and progressive updating. First, the task state bootstrapper constructs an initial task state from the task description and the relevant OpenAPI schemas. The task state updater then progressively revises this state given tool responses.

The main prompt for the task state bootstrapper is as below.

You are initializing the system state for a task execution system with multiple toolkits based on the given
↪ background information.
IMPORTANT: System state should contain ONLY these two types of data:
1. **Domain Data (Databases)**: The actual data that tools operate on
  * FileSystem toolkit: files, directories structure
  * Airline toolkit: users, flights, tickets, bookings
  * Message toolkit: messages, inbox items
  * These are stored at appropriate top–level or domain–specific keys
2. **Runtime Variables**: Execution context and session state
  * Store these DIRECTLY under 'runtime\_state' (flat structure)
  * Examples: current\_working\_directory, current\_user, is\_logged\_in, session\_token
  * IMPORTANT: Read toolkit descriptions carefully for initialization requirements

CRITICAL:
* NO 'runtime\_state.toolkits' structure – keep runtime\_state FLAT
* NO nested toolkit sections within runtime\_state
* NO duplicate concepts (e.g., only ONE current\_directory for the whole system)
* NO static values, validation rules, or schema metadata

Example of CORRECT runtime\_state structure:
"runtime\_state": {{
"current\_working\_directory": "/root"
}}

Rules:
1. Preserve all existing structures in the backgound information
2. Add runtime variables DIRECTLY under 'runtime\_state' (flat structure)
3. Add domain data at appropriate keys (not in runtime\_state)
4. NEVER create 'runtime\_state.toolkits' or any similar nesting
5. Avoid duplicating the same concept
6. Output valid JSON only

Background information:
{background_information}

Toolkits summary:
{json.dumps(toolkits\_summary, indent=2)}

Return the UPDATED config JSON with necessary domain data and runtime state.

The main prompt for the task state updater is as below.

You are an expert at tracking the execution state of a task.
Update the system state based on the tool calls and their effects on the system.

IMPORTANT GUIDELINES
1. **State Tracking Principles**
   – Update the system state to reflect ALL persistent state changes caused by tool calls
   – Operations that create, modify, or delete resources MUST update the corresponding structures
   – {"In synthesis mode Store ALL synthesized data from read operations as ground truth state" if synthesis_mode
   ↪ else "Operations that just query or read data should NOT add their results to the system state"}
2. **system state Organization**
   – When tool operations modify existing structures, update them directly (e.g., adding a new directory should add it
   ↪ to the directory tree)
   – For execution context that doesn't fit existing domain structures, use the root–level "runtime_state"
   – The "runtime_state" section is ONLY for execution context and ephemeral telemetry (e.g., current
   ↪ location/cursor, active selections, session info, temporary counters)
   – DO NOT store canonical domain data in "runtime_state" (e.g., files, inbox messages, database rows must live
   ↪ under their domain keys)
   – If a counter already lives under "runtime_state" (e.g., runtime_state.toolkits.messageapi.message_count), update
   ↪ it ONLY when the tool call semantics deterministically imply the change; never infer from read–only calls
   – Never duplicate the same fact both in a domain section and in "runtime_state"
3. **Value Formatting**
   – When recording locations, positions, or identifiers, use complete, unambiguous values
   – Avoid partial or relative references that could be misinterpreted
   – Preserve the format conventions used in the original system state
4. **What Changes to Track**
   – Resource creation/deletion/modification (files, directories, database records, etc.)
   – State transitions (status changes, position changes, mode switches)
   – Context updates (current location, active selections, session data)
   – DO NOT track query results, search results, temporary computations, or read–only operation outputs
5. **Example Structure with runtime_state**
   ```
   {{
     "DomainSystem" {{
       // Domain resources with any modifications from state–changing operations
     }},
     "runtime_state" {{
       // Execution context only (no canonical domain data)
       "current_context" "...",
       "current_user" "USR001"
     }}
   }}
   ```

Output the updated system state in JSON format only.

# F. Task Feedback Generation Implementation Details

Task feedback generation has two steps: checklist generation and judgement generation.

## F.1. Checklist generation

Given the task description, conversation history, and optional rules for the planning agent, we generate a detailed checklist that decomposes and clarifies the task intent into clear and verifiable objectives.

The main prompt for checklist generation is given below.

---

You produce ATOMIC, STATE−OPERATION−BASED verification checklists for tasks in ANY domain.
PREVIOUS TASKS (assumed done; resolve references only, do NOT re−verify): {prev_text}
CURRENT TASK: {current_task}

CRITICAL MULTI−TURN CONTEXT RULE:
When task mentions "values obtained", "results from previous", or specific counts like "three values", these refer to
↪ OUTPUT from the LAST task in PREVIOUS TASKS, not data from earlier tasks

RULES
1) Verify ONLY the current task. Return the MINIMAL set; if one item suffices, return EXACTLY one.
2) Each item = one pass/fail assertion about final state (implied by the question, do not guess the answer by yourself)
↪ or an executed operation (no procedures).
3) IDENTIFY ALL SEMANTIC UNITS: Each complete thought, question (direct or indirect like "I wonder"), or
↪ action in the task needs verification
4) PRESERVE LOGICAL FLOW: When actions depend on prior information or results, verify each step
5) Use explicit identifiers/paths/IDs when inferable; avoid vague pronouns.
6) RESOLVE AMBIGUOUS REFERENCES: When the current task contains pronouns like "the file", "it", "that
↪ item", etc., resolve them to specific entities based on PREVIOUS TASKS context.
7) Do NOT add optional behaviors (saving/exporting/logging/formatting) unless explicitly required.
8) Search/lookup/filter. Assert the search was executed with the specified term/criteria; do NOT require matches
↪ unless asked.
9) Transform/update. Assert the stated post−condition holds; do NOT invent extra artifacts.
10) Copy operations. Verify: source file remains intact (copy preserves original); destination file exists with the new
↪ name.
11) Create/Delete to assert existence/absence as specified.
12) Discrete relocation between containers (domain−agnostic). If applicable and implied: destination container
↪ exists (if mentioned); entity absent at source (for move, not copy); entity present at destination.

OUTPUT
Return ONLY a pure JSON array of objects with a single key "description". No extra text.

---

## F.2. Judgement generation

The checklist is verified item by item by an LLM-based judge, which receives the checklist, current task state, executed tool calls, the response from the planning LLM, conversation history, and the corresponding OpenAPI schemas. The judge aggregates any checklist objectives that fail verification into a compact task-level feedback message, which is returned to the planning agent to guide subsequent refinements.

The main prompt for the LLM judge is as follows.

You are an expert in verifying a checklist based on the execution results.
You have access to:
1. Current system state: The system state after execution
2. Tool calls: The list of functions that were called WITH their results
3. Agent response: The agent's output/explanation (if available)
4. Conversation history: Previous turns showing the context of how data was obtained{history_text}

IMPORTANT: Some operations (like grep, sort, find, ls) are query operations that don't modify the system state.
For these operations, verify their execution by checking if the corresponding tool was called in the tool_calls list OR
↪ Looking for evidence in the agent_response (if provided) that the operation was performed and results were
↪ obtained

Guidelines:
1. Verify items in the checklist one by one
2. For state−modifying operations (mkdir, create file, cd), check the system state for changes
3. For query operations (grep, sort, ls, find), check tool_calls and agent_response
4. For efficiency−related checklist items, analyze whether multiple tool calls could be merged based on the tool
↪ definitions provided
5. Status should be one of ["success", "failed", "unknown"]
"success": Task completed (evidence in tool_calls/system state/agent_response)
"failed": Task NOT completed AND agent provided NO explanation
"unknown": Task NOT completed BUT agent explained why (e.g., "missing information", "need user confirmation",
↪ "tool unavailable")
6. Just modify the status and reasoning fields of the checklist items, do not include any other text outside the
↪ json.{tool_defs_text}
7. If a tool call is made as the task required, do not mark it as failed even if the result is not as you expected
8. Use conversation history to understand data references.
9. When evaluating relevance, consider the full multi−turn context to understand where numbers/data come from

Evaluation principles (keep these high−level and tool−agnostic):
Only mark "failed" when there is clear evidence that the requirement was not met and no explanation was provided
↪ by the agent; if evidence is incomplete or you are not sure, use "unknown".
Accept equivalent pipelines that produce the required final outcome, regardless of operation order or scope.
Do not fail solely because an intermediate step operated on a broader scope; fail only if the final required
↪ subset/result is missing or incorrect.

The output should be in json format:
[
    {{"name": "...", "description": "...", "reasoning": "...", "status": "success" or "failed" or "unknown"}}
    {{"name": "...", "description": "...", "reasoning": "...", "status": "success" or "failed" or "unknown"}}
    ...
]
Do not include any other text outside the json.

# G. LLM-based API Schema Converter Implementation Details

The main prompt for the API schema converter is given below.

---

Convert this tool description to an OpenAPI 3.1 endpoint specification

Tool description:
{tool_description}

Create an endpoint object with these exact fields:
1. operationId: {tool['name']}}
2. summary: one−line description (keep it short)
3. description: brief description of the operation (1 sentence max)
4. requestBody: proper schema based on parameters
5. responses: ONLY USE STATUS 200 with oneOf schema for success/error
   – Success response: Based on tool's PURPOSE (not generic "result")
   – Error response: Standard error object
6. Analyze the tool's PURPOSE to generate an appropriate response schema. For example:
If it calculates something (area, factorial, etc.), return the calculated value
If it fetches data (user info, list of items), return the data structure
If it performs an action (create, update, delete), return success confirmation with relevant details
If it searches/filters, return matching results

CRITICAL REQUIREMENTS:
– NEVER use $ref − always inline all schemas
– ONLY use HTTP status 200 for ALL responses
– Use oneOf schema in the 200 response to handle both success and error cases
– All properties MUST have a "description" field
– The requestBody must have a schema with type "object"
– If no parameters, still include requestBody with empty properties

Return ONLY the endpoint object JSON.

---

# H. Detailed Inference Protocol of GATS

GATS uses Gecko as a pre-execution simulation environment for refining tool-call trajectories before final real-tool execution. For each task, the planning LLM uses Gecko's simulated tools to generate a candidate trajectory. Gecko returns validation feedback, simulated tool responses, task-state updates, and task-level feedback. If the task feedback indicates failure, the failed trajectory and feedback are provided to the planning LLM for another retry. Each retry starts from the same initial Gecko state snapshot, so simulated tool calls from failed attempts do not affect later attempts.

If Gecko judges a simulated trajectory as successful, GATS uses this trajectory as in-context guidance for the final assistant. The simulated trajectory is not directly used as the final answer. Instead, the final assistant still solves the task in the benchmark environment using real tools, and is instructed to verify and update any simulated or outdated values through its own tool calls. If no simulated attempt succeeds within the retry budget, no guidance is added and the method reduces to the original baseline setting.

# I. When to call real tools or simulated tools for $\tau^2$-bench

As described in 6.1, $\tau$2-bench does not provide databases as BFCLv3 does. Therefore, we use real tools to query real databases when information is needed. In our implementation, we classify the tools in $\tau$2-bench into 'readable' and 'writable' categories. Tools in the former class query real databases, provide real information, and do not take impactful actions. Examples include get_order_details. In contrast, tools in the latter category may use the information provided by read-only

*Table 6.* All-mock versus hybrid GATS on $\tau^2$-bench with GPT-5-mini.

| Method | $\tau^2$-retail | $\tau^2$-airline | Overall |
|---|---|---|---|
| GPT-5-mini | 73.5% | 57.0% | 65.3% |
| +GATS (all mock) | 75.2% | 59.5% | 67.4% |
| +GATS (hybrid) | **78.5%** | **65.0%** | **71.8%** |

*Table 7.* Error types corrected by GATS in cases where the baseline fails but GATS succeeds. Categories are not mutually exclusive.

| Setting | Error Type | Count |
|---|---|---|
| Multi-turn | Incomplete plans | 16/28 |
| Multi-turn | Wrong tool selection | 9/28 |
| Multi-turn | Wrong arguments | 12/28 |
| Multi-turn | Excessive trial-and-error | 1/28 |
| Single-turn | Missing/extra function calls | 10/19 |
| Single-turn | Incorrect argument content | 7/19 |
| Single-turn | Missing parameters | 2/19 |

tools and take actions that would make a difference. Examples include cancel_pending_order. When the planning LLM makes tool calls, we use real tools or simulated tools based on the categorization.

## J. All-mock and Hybrid GATS on $\tau^2$-bench

$\tau^2$-bench relies on hidden domain databases, such as user orders, reservations, and available flights. If Gecko simulates read-only database-query tools, it must also simulate these hidden databases, which can introduce simulation–reality drift. Therefore, our main $\tau^2$-bench experiments use a hybrid setting: read-only tools are executed against the real benchmark database, while state-changing tools are simulated in Gecko. Read-only tools provide information but do not modify the database state, whereas state-changing tools may update orders, reservations, or user records. In $\tau^2$-retail, 6 out of 13 tools are read-only; in $\tau^2$-airline, 6 out of 12 tools are read-only.

To isolate Gecko's contribution, we also evaluate an all-mock variant that simulates both read-only and state-changing tools. As shown in Table 6, all-mock GATS still improves GPT-5-mini over the baseline, while the hybrid setting further improves performance by reducing drift from hidden database queries. This indicates that the gains do not come solely from real read-only execution, and that hybrid execution is a useful deployment strategy when exact database information is needed.

## K. Error-type Analysis

To better understand where GATS improves tool use, we analyze 47 cases where the baseline fails but GATS succeeds, including 28 BFCLv3 multi-turn cases and 19 BFCLv3 single-turn cases. The error categories are not mutually exclusive. As shown in Table 7, in multi-turn tasks, GATS most often corrects incomplete plans, wrong tool selection, and wrong arguments. In single-turn tasks, GATS mainly corrects missing or extra function calls, incorrect argument contents, and missing parameters. These results suggest that GATS is not merely retrying until success; Gecko's feedback helps correct multiple types of tool-use errors.

## L. Experiments on the accuracy of argument validation

We evaluated the argument validator on the BFCL-Live-Simple subset. GPT-4o was used to generate tool calls for each example. For tool calls that passed the official BFCLv3 evaluation, we marked them as correct. For tool calls that failed the BFCLv3 evaluation, human annotators labeled each failure as either a *syntactic error* (violates the tool schema, e.g., missing required field or wrong field name/type) or a *semantic error* (schema-conformant but semantically inappropriate, e.g., wrong granularity or implausible value). These labels formed the ground truth.

We then ran our argument validator on the same generated tool calls and compared its predictions (correct / syntactic error/ semantic error) to the ground truth. Table 1 reports the detection rates.

