# OpenReview forum: "Gecko: A Simulation Environment with Stateful Feedback for Refining Agent Tool Calls"
_ICML.cc/2026/Conference — ICML 2026 regular_

### Official Review · Reviewer_8ZGc · 2026-03-09

**Soundness:** 2
**Presentation:** 1
**Significance:** 1
**Originality:** 3
**Overall Recommendation:** 3
**Confidence:** 5

**Summary:**

Tool calling ability is important to LLM agents. However, tool calling from LLM is prone to error because it relies on the LLM's intrinsic capabilities. Iteratively refining the tool call from execution results is expensive and unsafe. This paper proposes Gecko, an environment that simulates tool responses using a combination of rules and LLMs. Gecko synthesizes results that conform to input and output requirements, which allows effective iterative refinement for LLMs. Gecko improves the existing LLM performance on BFCL and $\tau^2$ bench.

**Compliance With Llm Reviewing Policy:**

Affirmed.

**Final Justification:**

I raised my score from 1 to 3 after communicating with the authors in the rebuttal period. My final verdict can be seen in my last "official comment" to the authors. Gecko itself is useful in certain use cases where the simulation of tool call results does not have strict requirements on the accuracy of the simulation. The rebuttals have changed my mind on several points I raised in the initial review, which made me updated my score. I am keeping the score to weak reject, due to my main concern over the accuracy in the simulation.

**Key Questions For Authors:**

1. In section 3.1, why does the paper use an LLM to check things that can and should be statically checked? All the examples listed in the section, such as countries, date formats, and ages, can and *should* be statically verified, because there are well-defined schemas and ground truths for those fields. Why use an LLM? Please show other "semantic" examples for the Argument Validator, otherwise this section can be replaced by a simple description of a rule-based checker.
2. One of Gecko's motivations is to reduce "unsafe" output. How does the paper make sure Gecko's simulated responses themselves are correct and safe to the API, *beyond simple schema checks?*
3. In section 3.4, how does Gecko ensure that the safety checklist is comprehensive for the tool call? Does Gecko requires user to input the checklist? How does the author measure Gecko's comprehensiveness in generating the checklist?
4. In section 5's comparison with STB, the paper argues that STB is too expensive because STB requires the collection of real responses from the API. It can also be argued that STB's approach is way more reliable because the tool call response is *grounded* by the real API execution. Again, how does Gecko make sure that the generated API responses are at least close to, if not exactly is, the real API response's data distribution?

**Limitations:**

Yes

**Strengths And Weaknesses:**

### Strengths
- Tool calling is an important problem for LLMs.
- Gecko does not require real API execution for tool call response simulation, reducing cost.

### Weaknesses
- The argument against using real tool call response to improve tool call capability can be further improved, i.e., it is not a complete deal breaker to execute real tool calls. Use cases that warrant executing real tool calls often have strict requirements on the fidelity and quality of the response. Moreover, if the tool call is stateless, then executing it will not incur side effects at all. If the paper wants to argue that executing a real tool call is untenable or infeasible, then the example and evaluation should focus on such cases. In addition, it is highly unlikely that in real-life, LLM users will blindly execute tool calls returned by the LLM and implement system-level domain-specific checkers. This renders Gecko's argument validator completely obsolete.
   - In essence, if Gecko simulates the API responses only based on API descriptions and output schema, then its effectiveness in real-life, where nuances exist *beyond* these descriptions, is questionable.
- It is unclear how well Gecko's response generator generalizes to other domains, or *beyond benchmarks*. It is fundamentally hard to simulate a correct response when it requires domain expertise to do so. In an extreme case, imagine a tool call that attempts to allocate a new page in memory in the operating system kernel. How can Gecko correctly simulate such a response?
- The paper has multiple typos and colloquial expressions. Its presentation can be greatly improved.

---

> ### Author Rebuttal · Authors · 2026-03-31
>
> We thank reviewer 8ZGc for the thoughtful comments. We first apologize that Sec. 6.1 did not clearly explain how GATS is used during inference. We will revise Sec. 6.1 and the introduction to make this explicit. Concretely, inference involves **two roles** that use the **same underlying base model**:
>
> (1) a **planning LLM inside GATS**, which interacts with Gecko’s simulated tools to produce a **mock solution**;
>
> (2) the **assistant LLM evaluated on the benchmark**, which interacts with the benchmark’s **real tools** to solve the task.
>
> The mock solution is **not** evaluated directly. Instead, if Gecko’s judge marks it successful, we prepend it to the user message as an in-context example, together with the instruction: *“The given plan could solve the task. Follow it in principle. Since the plan may contain outdated values, you should update it using results from your own tool calls.”* The assistant LLM then solves the task using the **real tools**. If all mock attempts fail, no example is provided, and the assistant LLM reduces to the original baseline.
>
> **Q1. Why use an LLM in the argument validator?**
> In many tools, important constraints appear only in natural-language descriptions. Examples:
>
> (1) `find_nearby(location: string)` in BFCL `simple_35`: the description requires *“City, State”*; `Chicago, IL` is valid, but `Chicago, US`  is not.
>
> (2) `find_athlete(name: string)` in BFCL `simple_311`: the description requires the **full name**; `LeBron James` is valid, but `James` is not.
>
> (3) `touch(file_name: string)` in BFCL `multi_turn_base_0`: the description specifies that the file name is local to the current directory and **must not contain a path**.
>
> Thus, LLM validation complements rule-based validation when semantics are specified only in text.
>
> **Q2. How does Gecko reduce unsafe behavior?**
> A key benefit is avoiding harmful trial-and-error on real tools. For example, in BFCL `multi_turn_base_38`, the user asks to remove the file named `report` in a specific project. The LLM's initial attempt may mistakenly delete a different file with the same name. With real-tool refinement methods (e.g., Self-Refine), the agent may detect the error and correct it subsequently, but the wrongly deleted file is still gone. In contrast, GATS explores in Gecko’s virtual environment; after a failed mock attempt, the environment resets to the initial task state. This allows learning from failure **without carrying over harmful side effects**.
>
> **Q3. How is the checklist produced and validated?**
> The checklist is generated by the judge LLM and is **not** provided by the user. The judge then evaluates whether the proposed solution and Gecko’s task state satisfy all checklist items. We treat the checklist as part of judge feedback, and its usefulness is supported by the ablation on judge feedback in Fig.~4(a).
>
> **Q4. Why is Gecko better than STB?**
> API outputs are not free-form samples; they are determined by the request arguments and backend state/runtime. Thus, cached real responses in STB may help with format, but they do not guarantee the accuracy of the concrete returned values for a new request as well. Gecko is more behaviorally faithful in two important respects: 1. Gecko validates tool calls before response generation, so invalid requests can be rejected rather than fabricated; 2. Gecko generates responses conditioned on a session-level runtime state, which enforces cross-call consistency. For example, if an agent first calls `create_event(title="meeting", date="2026-01-01", time="9:00")` and then calls `list_events(date="2026-01-01")`, the second response generated by Gecko will include the newly created event. By contrast, STB does not explicitly maintain such session-level state, making it less suitable for simulating multi-step dependencies.
>
> **W1. On using real tool calls.**
> We are **not** against using real tool calls. Our assistant LLM still solves the task using the **real tools**. Our core motivation is to avoid **trial-and-error on real tools** during refinement. We provide a concrete example in Q2. We agree with the reviewer that **stateless tool calls do not incur side effects**. In fact, in our (\tau^2)-bench setup (Sec.~6.1), we use **real execution for read-only tools** and **mock execution for state-changing tools**. This hybrid design reduces Gecko’s burden of simulating pure read-only tools while still preserving the main benefit of safe trial-and-error for actions with side effects.
>
> **W2. On realism and generalization of the simulation.**
> We agree that in specialized domains Gecko may fall short of domain expertise. However, Gecko is **not** intended to provide perfectly accurate simulation for every possible task. Its purpose is to provide a simulated environment in which an LLM can preview likely execution outcomes before solving the task with real tools and thereby avoid some mistakes in advance.
>
> **W3. On presentation.**
> We will revise the paper to fix typos and colloquial expressions.

---

> > ### Author Rebuttal · Reviewer_8ZGc · 2026-04-01
> >
> > Thank you authors for the rebuttals.
> >
> > It is unfortunate that the submission is not very clear in its description of GATS. The paper might need much changes to improve the presentation so that such ambiguity does not appear anymore.
> >
> > Regarding Q1's response, in examples (1) and (3) you gave, again, they both can and should be checked statically. There is a finite set of (City, State) pairs in the U.S., and the authors can easily enumerate through them. Regarding (3), it is hard to be convinced that an LLM, naturally stochastic, can correctly decide whether a given path is in the current working directory or not. Moreover, a posix path or a Windows path can _still_ be easily statically checked. Lastly, in example (2), there are people with only one word as their full name, such as Bhavya. I fail to see how can an LLM correctly validate against such cases, and the 3 examples you gave me really do not convince me. If the authors would like to argue that an LLM is necessary in judging whether inputs to a function is correct, then they need to find a case, where (1) the static checkers cannot capture the _semantic meaning_ of the input arguments, and (2) that said semantic meanings _cannot_ be statically expressed, and (3) that said semantic meaning is necessary for the correctness of the function call. Sadly, the given examples do not satisfy these criteria.
> >
> > Regarding Q2's response, one can achieve the same effect through a lightweight file system wrapper that backs up the target file before execution (a simple `cp` command suffices), which will give both (1) recoverability and (2) authentic function call response. What's Gecko's advantage over this? Granted, the authors can argue that Gecko is even _lighter_ weight because it doesn't execute the actual tool call at all, but one could still argue that this boils down to the trade-off of tool call execution cost versus the authenticity of the simulated response.
> >
> > Regarding Q4's response. To the point of "Gecko generates responses conditioned on a session-level runtime state, which enforces cross-call consistency," this means that Gecko _does_ maintain an internal state to serve stateful function calls. This brings two questions: (1) how does Gecko knows _how_ to store the states that satisfy the tool call's semantic? Dates can be easy, what about memory addresses? (2) how does Gecko knows _how_ to serve/simulate the stateful tool calls? Returning dates can be easily, what about other long-horizon tasks?
> >
> > All in all, unfortunately, in this paper and the rebuttals, the authors did not fully convince me that saving the cost of tool call execution is worthy of risking inaccurate simulation results. I understand that Gecko is not meant to replace real execution of tool calls, but even considering the simulation nature, Gecko, or the submission, does not make a convincing case that the simulation is good enough to replace the real execution results.

---

> > > ### Author Response · Authors · 2026-04-01
> > >
> > > **-----4.7 Update-----**
> > >
> > > Thank you again for the thoughtful follow-up and for the constructive discussion throughout the rebuttal process. We are very glad that our responses helped address several of your earlier concerns, and we sincerely appreciate that you raised your overall recommendation to 3.
> > >
> > > Regarding your remaining concern on **simulation accuracy**, we would like to address it below.
> > >
> > > We agree that simulation accuracy is an important consideration, and we also agree that simulation is inherently imperfect relative to real execution, as discussed in **Sec. 5 Limitation**. Our claim is **not** that Gecko provides perfect simulation fidelity; instead, it is the narrower claim that **Gecko’s informative simulation is sufficient to support tool-call refinement across the benchmarks and models we evaluate**.
> > >
> > > **Table 2 provides evidence for this point.** When simulated tool executions are used in place of real ones, the additional degradation is **limited rather than overwhelming** in the BFCL setting: under **real tools + LLM judge**, the judge achieves **96.7% TP** and **87.5% TN**; replacing real tools with simulated tools changes this to **91.3% TP** and **87.5% TN**. A more detailed explanation is provided in **Rebuttal to Reviewer ZC4u Q1**.
> > >
> > > We also believe the **$\tau^2$-bench all-mock ablation** is highly relevant (see rebuttal to ZC4u Q2), because Gecko cannot accurately simulate **database-dependent** tools. Even in this harder setting, **+GATS (all mock)** still improves over the baseline rather than degrading: **65.3% → 67.4%**, while **+GATS (hybrid)** reaches **71.8%**.
> > >
> > > We view this as additional evidence that Gecko’s refinement signal remains useful even when simulation is imperfect and the domain is harder to simulate faithfully. A plausible reason is that: if the **judge marks all mock solutions unsuccessful**, then **no in-context example is provided**, and the assistant LLM falls back to the baseline setting and solves the task through real tools. In addition, Gecko **recalibrates** its estimated task state using **real tool-call results** after each turn, which can better ground later simulation in longer-horizon tasks such as those in $\tau^2$-bench (**see Sec. 5 Do simulation errors accumulate?**).
> > >
> > > Refining in simulation avoids external side effects and the cost of executing real tools, but simulation is inherently imperfect. This makes refinement in simulation a harder setting than refinement on real tools. Gecko makes this harder setting effective in practice: Table 1 reports **+5.6%** averaged over 8 models on BFCLv3, and Table 3 reports **+5.1%** averaged over 4 models on $\tau^2$-bench. **Precisely because simulation is inherently imperfect relative to real execution, these gains are nontrivial.**
> > >
> > > We hope this helps clarify how we view the role of simulation in Gecko: not as a perfect replica of real execution, but as a sufficiently informative environment for pre-execution refinement.
> > >
> > > **-----Previous-----**
> > >
> > > **Q1: static vs. LLM validation**
> > >
> > > Our earlier examples showed constraints that appear only in free-form descriptions without formal specifications, so **no universal static checker** can cover them all. The reviewer’s point appears to be that, for each individual tool, a custom static checker could in principle be written. However, this assumes tools are **known in advance**. In BFCLv3 and real-world deployment, tool sets are open-ended and may only be exposed at inference time. Writing per-tool static validators does not scale.
> > >
> > > **We do not claim LLM validation is universally superior to static checkers.** As described in Sec. 3.1, Gecko already uses rule-based validation for all schema-expressible constraints. The two are **complementary**: rule-based for formally specified constraints, LLM-based for those that appear only in natural-language descriptions.
> > >
> > > **Q2: per-tool workarounds**
> > >
> > > The existence of workarounds for a few specific tools does **not reduce the value of Gecko’s simulation**. Gecko **avoids external side effects during refinement** because it does not execute real tools, and because it is built from tool definitions rather than additional tool-specific engineering, it is inherently **general and scalable**.
> > >
> > > **Q4: stateful simulation**
> > >
> > > The question assumes that serving stateful tool calls requires a hand-written, tool-specific state model. Gecko does not work that way. It does not require a predefined state schema or hand-coded state-transition rules for each tool. Instead, as described in Sec. 3.2 and Sec. 3.3, after each tool call, Gecko uses the LLM to update a **task-relevant state summary** from the trajectory so far, and subsequent response generation is conditioned on that state together with the tool schema.
> > >
> > > **Presentation**
> > >
> > > The main point that may have been unclear is **how GATS’s mock solution is used**: it is prepended as an in-context example, not executed directly. We will revise Sec. 6.1 and Sec. 1 to make this explicit.

---

### Official Review · Reviewer_ZC4u · 2026-03-12

**Soundness:** 2
**Presentation:** 3
**Significance:** 3
**Originality:** 2
**Overall Recommendation:** 4
**Confidence:** 4

**Summary:**

This paper introduces a simulation environment of tool calling, namely Gecko, which replaces real tool execution during LLM inference-time refinement. Gecko provides three types of feedback: argument validation, simulated tool responses, and task-level completion feedback. Based on Gecko, a test-time scaling method GATS iteratively refines tool call sequences using these feedback signals. The experiments on BFCLv3 and τ²-bench demonstrate consistent improvements across GPT-4o, GPT-5, Gemini-3.0-pro, and other LLMs, with Gemini-3.0-pro+GATS achieving new SOTA on BFCLv3.

**Compliance With Llm Reviewing Policy:**

Affirmed.

**Final Justification:**

I raise my overall score to 4 since the authors' rebuttal addressing my concerns.

**Key Questions For Authors:**

1. The fidelity validation in Sec. 3.2 relies on ground truths derived from real tools + rule-based state matching on BFCL-Multi-Turn-Base. However, if rule-based matching itself has non-trivial error rates, the reported 5.4% gap between real and simulated tools may be unreliable. Can you report the accuracy of the rule-based state matching independently, and clarify whether noise in ground truth construction could systematically bias this comparison?
2. You utilized real read-only tools for $\tau^2$-bench due to simulation inaccuracies. Can you quantify exactly how much the performance gain is attributed to these real reads versus Gecko simulation? An ablation study replacing real reads with simulated ones (even if inaccurate) would clarify Gecko's isolated contribution.
3. You use the planning LLM as the judge. How do you prove this does not cause circular evaluation bias (the model validating its own hallucinations)?
4. Semantic argument validation only detects 60-70% of errors. How will you solve the simulation-reality drift when deploying this in complex real-world APIs?
5. The test-time scaling introduces significant latency overhead. Fig. 5 shows latency increases to ~18 seconds for 3 retries. This time cost is impractical for real-world applications. Do you have any strategy to optimize this inference overhead or dynamically early-stop the retry loop?

**Limitations:**

Yes

**Strengths And Weaknesses:**

Strengths:
1. Test-time scaling for agents without spending real API costs is effective and useful.
2. The authors have conducted extensive experiments on multiple LLMs (GPT-4o, Gemini, etc.), demonstrating the proposed method's superiority. The ablation studies clearly isolate and prove the necessity of each module.
3. The paper is well-structured and easy to follow. Fig. 2 and Fig. 3 effectively illustrate the system pipeline and feedback examples. The discussion section honestly acknowledges limitations and future directions.

Weaknesses:
1. The proposed method is too engineering-heavy, which is basically prompt engineering and API stitching. It lacks significant algorithmic novelty or theoretical depth.
2. Simulation fidelity is insufficiently validated. The only evidence is a binary success/failure comparison on a single dataset (Table 2), which cannot capture content-level response deviations. τ²-bench further sidesteps the hardest cases by using real tools for all read-only APIs. If simulation errors are systematic, GATS's gains may reflect adaptation to Gecko's biases rather than genuine improvement.
3. The source of improvement is unclear. It is unknown whether GATS gains come from correcting argument errors, fixing tool selection, or simply retrying. The ablation (Fig. 4a) is limited to multi-turn tasks and contains no error-type breakdown, leaving the mechanism behind GATS opaque.
4. Generalization scope is overstated. Gecko only supports text-output tools and cannot reliably simulate database-dependent tools. Combined with the τ²-bench real-tool fallback, the validated scope is considerably narrower than the "comprehensive environment" framing implies.

---

> ### Author Rebuttal · Authors · 2026-03-31
>
> We thank reviewer ZC4u for the detailed comments.  We first apologize that Sec. 6.1 did not clearly explain how GATS is used during inference. Inference involves **two roles** that use the **same base model**:
>
> (1) a **planning LLM inside GATS**, which interacts with Gecko’s simulated tools to produce a **mock solution**;
>
> (2) the **assistant LLM evaluated on the benchmark**, which interacts with the benchmark’s **real tools** to solve the task.
>
> The mock solution is **not** evaluated directly. Instead, if Gecko’s judge marks it successful, we prepend it to the user message as an in-context example, together with the instruction: *“The given plan could solve the task. Follow it in principle. Since the plan may contain outdated values, you should update it using results from your own tool calls.”* The assistant LLM then solves the task using the **real tools**. If all mock attempts fail, no example is provided, and the assistant LLM reduces to the original baseline.
>
> **Q1/W2. On simulation fidelity and its validation.**  We apologize that Sec. 3.2 was unclear. In this experiment, the ground-truth labels are **not** constructed by rule-based state matching. Instead, we use the **BFCL evaluator** to label baseline GPT-4o solutions on BFCL-Multi-Turn-Base as success or failure. We then compare whether our LLM judge reaches the same decision under **real-tool** vs **simulated-tool** executions. The reference to “rule-based state matching” here was misleading and will be removed in revision. Under **real tools + LLM judge**, the judge achieves **96.7\% TP** and **87.5\% TN**, showing that the judge itself is reasonably reliable. Replacing real tools with simulated tools reduces TP to **91.3\%** while keeping TN at **87.5\%**. Thus, Table 2 jointly evaluates judge reliability and the additional degradation introduced by simulated tool responses.
>
> **Q2/W4. The hybrid $\tau^2$-bench design, and Gecko’s isolated contribution.**
> Our claim is not that Gecko already provides a comprehensive simulator for arbitrary APIs or reliably models all database-dependent tools. Rather, our claim is that Gecko is useful for **pre-execution refinement** in the benchmark settings we evaluate on.
>
> We also performed the requested ablation on $\tau^2$-bench:
>
> | Method | $\tau^2$-retail | $\tau^2$-airline | Overall |
> |---|---:|---:|---:|
> | GPT-5-mini | 73.5% | 57.0% | 65.3% |
> | +GATS (all mock) | 75.2% | 59.5% | 67.4% |
> | +GATS (hybrid) | 78.5% | 65.0% | 71.8% |
>
> Even in the **all-mock** setting, GATS improves over the baseline, so the gain does not come solely from using real read-only tools.
>
> **Q3. On using the same backend LLM for planning and judging.**
> In our experiments, the planning LLM and the judge share the same backend LLM, but this is a **controlled experimental choice**, not a requirement of Gecko: the judge is fully configurable. We use the same backend model to avoid introducing an additional model confound, so that improvements can be attributed more cleanly to Gecko/GATS itself rather than differences between two separate models.
>
> **Q4. On simulation-reality drift.**
> We agree that 60--70% semantic-error detection is not enough to eliminate simulation-reality drift in complex real-world APIs. Our claim is therefore **not** that Gecko fully solves this problem. Instead, Gecko acts as a **pre-execution refinement layer** that can filter out a substantial fraction of invalid calls before real execution, which is already useful even if imperfect. For more complex deployments, stronger semantic validators, more domain-specific rules, and hybrid execution are natural ways to further reduce drift.
>
> **Q5. On latency.**
> We agree that test-time scaling introduces extra latency. In practice, GATS does not always exhaust the full retry budget: retries stop once the judge approves a mock solution. Moreover, Gecko’s internal agents are fully configurable, so deployment does not require using the same high-cost LLMs as in our experiments. Lower-cost backends can be used for some or all Gecko components to reduce inference overhead. Overall, this is a tunable **performance--latency trade-off**.
>
> **W3. On the source of improvement.**
> We conducted an error-type breakdown on GPT-4o’s BFCLv3 `multi_turn_base` and `non_live_simple` results, focusing on **47 cases** where the baseline failed but GATS succeeded (**28 multi-turn**, **19 single-turn**).
>
> | Setting | Error type | Count |
> |---|---|---:|
> | Multi-turn | Incomplete plans | 16/28 (57%) |
> | Multi-turn | Wrong tool selection | 9/28 (32%) |
> | Multi-turn | Wrong arguments | 12/28 (43%) |
> | Multi-turn | Excessive trial-and-error | 1/28 (4%) |
> | Single-turn | Missing/extra function calls | 10/19 (53%) |
> | Single-turn | Incorrect argument content | 7/19 (37%) |
> | Single-turn | Missing parameters | 2/19 (11%) |
>
> These results show that GATS corrects multiple error types.

---

> > ### Author Rebuttal · Reviewer_ZC4u · 2026-04-02
> >
> > Thanks a lot for your response addressing my concerns. I raise my overall score to 4.

---

> > > ### Author Response · Authors · 2026-04-03
> > >
> > > Dear Reviewer ZC4u,
> > >
> > > Thank you very much for your positive rating on our submission! We are glad that our clarifications fully addressed your questions, and we sincerely appreciate your time, effort, and support of our work!
> > >
> > > Best regards,
> > >
> > > Authors of Gecko

---

### Official Review · Reviewer_FKBA · 2026-03-22

**Soundness:** 3
**Presentation:** 2
**Significance:** 2
**Originality:** 2
**Overall Recommendation:** 4
**Confidence:** 4

**Summary:**

This paper presents an approach for using LLMs to simulate the execution of tool calls: the responses from the tool execution, and the change made to the environmental state. This allows an inference-time approach for improving LLMs' abilities to call tools correctly: a tool call is proposed, its results and task state are simulated, an LLM judges whether these results & task state are correct, and if not another tool call is proposed. The paper shows that this improves performance of a number of models on BCFL and subsets of \tau^2 bench.

**Compliance With Llm Reviewing Policy:**

Affirmed.

**Final Justification:**

The rebuttal addressed my weaknesses W3-W5. W1 (thin contribution/novelty) and W2 (experiments being somewhat scattered/incomplete) are still an issue, but I have updated my score to a 4.

**Key Questions For Authors:**

Questions
- Q1) What LLMs are used to implement each component of Gecko, in each experiment? Fig 4b gives some indicator for this (I think it's GPT-4.1-nano for the arg validator, and GPT-4o for the others?) but it should really be in the main text.
- Q2) What are the experiments in Table 2? I didn't really understand this from the description in the caption or in Sec 3.3 or 3.4.
- Q3)  What's being used to judge correctness in Table 2?
- Q4) How many iterations of refinement are being done in GATS?
- Q5)  Can you give more details about the alternate inference time methods in 6.3? e.g. what temperatures are being used in Best-of-n and how were they tuned; what is being used to choose from the candidates in best-of-n? How many iterations of refinement in reflexion? How is self-refine working? There's not enough info here to be convincing that these baselines are reasonable.

**Limitations:**

yes

**Strengths And Weaknesses:**

Strengths
- S1) The approach is somewhat simple, just requiring prompting models to predict tool call outputs and act as a judge of these outputs.
- S2) The approach shows consistent and what seem to be pretty substantial improvements across the evaluations.
- S3) The paper evaluates on multiple datasets, with a number of models (although see below for some comments about consistency).

Weaknesses
- W1) The contribution seemed a bit thin and I didn't feel there was too much novelty, as each of the individual components in the system is pretty straightforward or has been explored in previous work.
- W2) The experiments seemed incomplete, and the choice of models/datasets for each experiment seemed a bit haphazard:
- W2a) The paper only compares to alternate inference methods on one subset of a dataset (\tau^2-airline) and only for a single model (GPT-5-min).
- W2b) I appreciated that the paper presents ablations (Fig 4), but these are done on a different dataset (BFCL-Multi-Turn-Base) and for a different model (GPT-4o).
- W2c) The paper presents reference results from Claude Opus/Sonnet 4 and Kimi-K2-Instruct but doesn't apply the GATS method to them.
- W3c) The paper should present a majority voting baseline too.
- W4) The paper could better motivate why tool calls need to be simulated rather than just executed in the environment (I can imagine some motivations, like tool calls might have destructive effects, and would require resetting environments, are those right?)
- W5) Some details of the method and component evaluation were unclear, see questions below.

---

> ### Author Rebuttal · Authors · 2026-03-31
>
> We thank reviewer FKBA for the thoughtful comments. We first apologize that Sec. 6.1 did not clearly explain how GATS is used during inference. We will revise Sec. 6.1 and the introduction to make this explicit. Inference involves **two roles** that use the **same base model**:
>
> (1) a **planning LLM inside GATS**, which interacts with Gecko’s simulated tools to produce a **mock solution**;
>
> (2) the **assistant LLM evaluated on the benchmark**, which interacts with the benchmark’s **real tools** to solve the task.
>
> The mock solution is **not** evaluated directly. Instead, if Gecko’s judge marks it successful, we prepend it to the user message as an in-context example, together with the instruction: *“The given plan could solve the task. Follow it in principle. Since the plan may contain outdated values, you should update it using results from your own tool calls.”* The assistant LLM then solves the task using the **real tools**. If all mock attempts fail, no example is provided, and the assistant LLM reduces to the original baseline.
>
> **Q1. What LLMs are used?**
> In our experiments, the **argument validator** is fixed to **GPT-4.1-nano** to reduce overhead. All other Gecko components, including the **response generator**, **task state estimator**, **task feedback generator**, and the **planning LLM in GATS**, use the **same specified backend model** as the evaluated baseline assistant. This design is to avoid introducing a stronger model and ensures that the comparison reflects the effect of Gecko/GATS rather than a model upgrade. We will clarify this explicitly in the revision.
>
> **Q2/Q3. Explain the experiments in Table 2.**
> We apologize that Sec. 3.2/3.3 was unclear. In Table 2, the ground-truth labels are **not** constructed by our rule-based state matching. Instead, we use the **BFCL evaluator** to label sampled GPT-4o solutions on BFCL-Multi-Turn-Base as success or failure. We then ask our **LLM judge** to determine whether each solution succeeds when conditioned on either **real-tool executions** or **simulated-tool executions**, and compare the judge’s decision against the BFCL evaluator label.
>
> Therefore, Table 2 evaluates two things jointly:
> (1) how reliable the **judge** is under **real-tool executions**, and
> (2) how much additional degradation is introduced when **real tools are replaced by simulated tools**.
>
> Under **real tools + LLM judge**, the judge achieves **96.7% TP** and **87.5% TN**, showing that it is reasonably reliable at identifying both correct and incorrect solutions. Replacing real tools with simulated tools reduces TP to **91.3%** while keeping TN at **87.5%**. Thus, the **5.4%** gap should be interpreted as the loss introduced by simulated tool responses. We will revise the paper to make this experiment setup explicit and remove the misleading reference to rule-based state matching in this context.
>
> **Q4. How many iterations of refinement are being done in GATS?**
> The maximum number of refinement iterations in GATS is **3**.
>
> **Q5&W5. Can you give more details about the alternate inference-time methods in Sec. 6.3?**
> We agree that Sec. 6.3 was too brief and will revise it to make the baselines explicit. All baselines use the same evaluation harness, base model (**GPT-4o**), tool schemas, multi-turn inputs, and real tool environment; only the inference-time control strategy differs.
>
> For **Best-of-N**, we use **N=3** and **temperature=0.7**. At each turn, we sample 3 candidate JSON action plans, use an judge model to select a `best_plan`, and then execute only the selected plan with real tools. For **Reflexion**, we use **one reflection step per turn** and keep up to the **5 most recent reflections** as persistent guidance. For **Self-Refine**, we use **one draft–refine–execute pass per turn** (not an unbounded refinement loop). For completeness, **Majority Voting** uses **temperature=0.7** and samples **5** candidate tool-calling actions per decision step, selecting the winner by exact tool-call majority vote before execution.
>
> We will add more implementation details and hyperparameters directly to Sec. 6.3 in the revision.
>
> **W3. Majority Voting baseline**
> | Method | Performance | Tokens (k) | Avg. Tool Calls |
> |---|---:|---:|---:|
> | **w/ Majority Voting (N=5)** | **59.0%** | **291.7** | **6.62** |
>
> **W4. Motivations**
> Yes, a key benefit is avoiding harmful trial-and-error on real tools. For example, in BFCL `multi_turn_base_38`, the user asks to remove the file named `report` in a specific project. The LLM's initial attempt may mistakenly delete a different file with the same name. With real-tool refinement methods (e.g., Self-Refine), the agent may detect the error and delete the correct file, but the wrongly deleted file is still gone. In contrast, GATS explores in Gecko’s virtual environment; after a failed mock attempt, the environment resets to the initial task state. This allows learning from failure **without carrying over harmful side effects**.

---

> > ### Author Rebuttal · Reviewer_FKBA · 2026-04-04
> >
> > Thanks for your response! It addressed my weaknesses W3-W5. W1 (novelty) and W2 (experiments being somewhat scattered/incomplete) are still an issue, but I have updated my score to a 4.

---

> > > ### Author Response · Authors · 2026-04-04
> > >
> > > Deer Reviewer FKBA,
> > >
> > > Thank you very much for your positive rating on our submission! We are glad that our clarifications addressed the W3-W5. We also apologize for not answering W1 and W2 in the previous rebuttal due to the 5000 characters limitation space. Here we want to do our best to answer the remaining weaknesses.
> > >
> > > ### W1 (novelty).
> > > We agree that the individual components of Gecko, such as validation, data synthesis, and feedback generation, have each been explored in prior work. The novelty we want to highlight is that **we use them in a different way and for a different purpose**.
> > >
> > > Our central question is whether an agent can **preview likely execution outcomes before actually solving the task** and use that preview to improve tool-use performance. This is similar to how people may first sketch out a few possible solutions on scratch paper before giving a final answer. Gecko is built for this purpose. It is a virtual environment with multiple LLM-based components and rule-based engineering so that an agent can simulate likely tool-execution outcomes and receive grounded feedback before using real tools.
> > >
> > > This matters because many existing test-time scaling or refinement methods work directly on real tools. Refining in simulation avoids external side effects and the cost of executing real tools. For example, in BFCL `multi_turn_base_38`, the user asks to remove the file named `report` in a specific project. An initial attempt may delete a different file with the same name. With real-tool refinement, the agent may later discover the mistake and delete the correct file, but the wrong file is already gone. In contrast, GATS explores in Gecko’s virtual environment, and after a failed mock attempt, the environment resets to the initial task state. This lets the agent learn from failure **without carrying over harmful side effects**.
> > >
> > > We believe this pre-execution simulation is the key advance over prior refinement methods.
> > >
> > > ### W2 (experiments being somewhat incomplete).
> > >
> > > **(a) Comparing GATS with alternate inference methods on more datasets / models.**
> > > Thank you for this suggestion. We additionally compare GATS with alternate inference-time methods on **BFCL-Multi-Turn-Base with GPT-4o**:
> > >
> > > | Method | Acc | Avg total tokens/task | Avg tool calls/task |
> > > |---|---:|---:|---:|
> > > | Baseline GPT-4o | 61.00% | 30,197 | 7.33 |
> > > | w/ Reflexion | 63.50% | 36,056 | 8.10 |
> > > | w/ Merge-to-one | 58.00% | 61,630 | 7.46 |
> > > | w/ Self-Refine | 64.50% | 76,864 | 16.22 |
> > > | w/ Best-of-n | 63.00% | 78,745 | 14.04 |
> > > | w/ Majority Voting | 66.00% | 129,256 | 8.67 |
> > > | w/ GATS | 72.00% | 108,745 | 6.69 |
> > >
> > > **(b) Inconsistent settings between alternate-inference comparison (Table 5) and ablations (Fig. 4).**
> > > Thank you for pointing this out. The new experiment in **W2(a)** is conducted on the **same dataset (BFCL-Multi-Turn-Base)** and with the **same model (GPT-4o)** as the ablations in **Fig. 4**. We hope this makes the interpretation of both experiments much clearer and the overall evaluation less scattered.
> > >
> > > **(c) Missing +GATS results on the reference models**
> > > Thank you for this suggestion. We have now applied GATS to additional strong models:
> > >
> > > | Model | $\tau^2$-retail | $\tau^2$-airline | Overall |
> > > |---|---:|---:|---:|
> > > | Claude Sonnet 4.5 | 85.5% | 69.5% | 77.5% |
> > > | +GATS | 88.1% | 74.0% | 81.0% |
> > > | Kimi-K2-Instruct | 70.6% | 56.5% | 63.6% |
> > > | +GATS | 75.2% | 62.0% | 68.6% |
> > >
> > > In the revision, we will remove the previous reference-only rows (Claude Opus/Sonnet 4 and Kimi-K2-Instruct) from Table 4 and merge the above results into that table.
> > >
> > > We hope we have addressed your concerns as fully as possible. If you have any remaining concerns, please feel free to update your rebuttal acknowledgement, and we will do our best to address them. We sincerely appreciate your time, effort, and support of our work.
> > >
> > > Best regards,
> > >
> > > Authors of Gecko

---

### Official Review · Reviewer_JLHB · 2026-03-24

**Soundness:** 3
**Presentation:** 3
**Significance:** 3
**Originality:** 3
**Overall Recommendation:** 4
**Confidence:** 2

**Summary:**

This paper presents a simulation environment for refining agentic frameworks that make use of external tools. The environment checks the validity of tool calls including input arguments and tool names, synthesizes reasonable responses, with the help of another LLM, that adhere to the output schema, and assesses whether all task objectives have been achieved. Experiments show that this environment provides useful feedback for test-time scaling of modern LLMs.

**Compliance With Llm Reviewing Policy:**

Affirmed.

**Final Justification:**

The rebuttal addressed the concerns. I was already positive and I maintain my score.

**Key Questions For Authors:**

Can you evaluate the environment on coding tasks using Claude? Do you observe any improvemets?

**Limitations:**

yes

**Strengths And Weaknesses:**

Pros:

* This is a well written and well engineered paper with a clear goal.
* The proposed environment leverages a clever  combination of rules and LLMs to perform teh tool emulation and validation.
* While there are many works already that provide feedback loops for LLMs based on real tool use, the paper proposes a simulation environment that uses a "helper" LLM for this purpose.

Cons:

* The proposed work is conceptually simple and the innovation is in terms of methodology rather than algorithms. There is another similar environment (STB) but the current work proposes some advancement.
* The environment relies on the strength of other LLMs which are used at various steps. As LLMs improve, it is very possible that next generations will enable implementations of Gecko just by prompting.

The authors provide additional results for coding that strengthen the paper.

---

> ### Author Rebuttal · Authors · 2026-03-31
>
> We thank reviewer JLHB for the thoughtful comments. We first apologize that Sec. 6.1 did not clearly explain how GATS is used during inference. We will revise Sec. 6.1 and the introduction to make this explicit. Inference involves **two roles** that use the **same base model**:
>
> (1) a **planning LLM inside GATS**, which interacts with Gecko’s simulated tools to produce a **mock solution**;
>
> (2) the **assistant LLM evaluated on the benchmark**, which interacts with the benchmark’s **real tools** to solve the task.
>
> The mock solution is **not** evaluated directly. Instead, if Gecko’s judge marks it successful, we prepend it to the user message as an in-context example, together with the instruction: *“The given plan could solve the task. Follow it in principle. Since the plan may contain outdated values, you should update it using results from your own tool calls.”* The assistant LLM then solves the task using the **real tools**. If all mock attempts fail, no example is provided, and the assistant LLM reduces to the original baseline.
>
> **W1. On novelty / contribution.**
> We agree that Gecko is primarily a **system/method contribution** rather than a new standalone learning algorithm. Its main contribution is to provide a **virtual sandbox** in which an agent can simulate likely tool-execution outcomes and receive grounded feedback **before real execution**. This is important because many existing test-time scaling/refinement methods operate directly on **real tools**, which can be risky when actions have side effects. For example, in BFCL `multi_turn_base_38`, the user asks to remove the file named `report` in a specific project. An initial attempt may mistakenly delete a different file with the same name. With real-tool refinement, the agent may later detect and correct the mistake, but the wrongly deleted file is already gone. In contrast, GATS explores in Gecko’s virtual environment; after a failed mock attempt, the environment resets to the initial task state. This allows learning from failure **without carrying over harmful side effects**. We believe this pre-execution sandboxing is the key practical advance over prior refinement methods.
>
> **W2. On reliance on other LLMs / possible obsolescence.**
> We agree that Gecko’s quality depends on the capability of the underlying LLMs. However, we do not view this as making Gecko obsolete; rather, it reflects the fact that Gecko is an **infrastructure layer** for tool-use agents. Table 5 already shows that even current strong LLMs cannot simply recover the same behavior by one-shot prompting: for example, **Merge-to-one** performs substantially worse than GATS. In addition, Gecko combines LLM components with **rule-based validation**, which remains useful even as models improve. More importantly, as discussed in Sec. 5 (“New research possibilities enabled by Gecko”), we view Gecko as a broader simulation environment for tool use, of which **GATS is only one application**. Stronger future LLMs may improve Gecko’s components, but the environment itself can still enable new tool-use training, evaluation, and refinement settings.
>
> **Q1. Can you evaluate the environment on coding tasks using Claude? Do you observe any improvements?**
> Our intended domain is **tool-use assistants for everyday tasks**, such as e-commerce customer support or personal assistants, rather than coding benchmarks. For coding tasks, Gecko-style simulation is generally less compelling because there are many alternative ways to obtain direct execution feedback, so the method would largely reduce to a more standard refinement pipeline. Nevertheless, to address the reviewer’s question, we applied GATS to **Claude Sonnet 4.5** on $\tau^2$-bench and still observe consistent gains:
>
> | Domain | Baseline | +GATS | Δ |
> |---|---:|---:|---:|
> | $\tau^2$-airline | 69.5% | 74.0% | +4.5% |
> | $\tau^2$-retail | 85.5% | 88.1% | +2.6% |
>
> These results show that GATS remains effective on a strong Claude model. We will clarify the intended scope of Gecko more explicitly in the revision.

---

> > ### Author Rebuttal · Reviewer_JLHB · 2026-04-03
> >
> > Thanks for the rebuttal. I am already positive and will maintain my score.

---

> > > ### Author Response · Authors · 2026-04-03
> > >
> > > Dear Reviewer JLHB,
> > >
> > > Thank you very much for your positive rating on our submission! We are glad that our clarifications fully addressed your questions, and we sincerely appreciate your time, effort, and support of our work!
> > >
> > > Best regards,
> > >
> > > Authors of Gecko

---

### Decision · Program_Chairs · 2026-04-30

**Decision:**

Accept (regular)

**Comment:**

This paper proposes Gecko, a simulated tool-use environment that allows LLM agents to refine tool calls before interacting with real tools, with the goal of reducing cost and unsafe execution. Reviewers generally agreed that the paper addresses an important practical problem and that the system is well-engineered and empirically effective across multiple benchmarks and models. The main remaining concerns are about novelty and scope. Multiple reviewers viewed the contribution primarily as a strong systems paper rather than a major algorithmic or theoretical advance, and one reviewer remained concerned that simulation fidelity may be insufficient for more complex or state-dependent tools. More broadly, the paper’s claims should be framed carefully. I lean towards acceptance, while encouraging the authors to sharpen the framing, clarify the limits of simulation fidelity, and better emphasize the method's intended scope.